# Generating variability from motor primitives during infant locomotor development

**Elodie Hinnekens[1,2]\*, Marianne Barbu-Roth[3], Manh-Cuong Do[1,2], Bastien Berret[1,2,4†], Caroline Teulier[1,2†]**

[1]Université Paris-Saclay, CIAMS, Orsay, France; [2]Université d'Orléans, CIAMS, Orléans, France; [3]Université de Paris, CNRS, Integrative Neuroscience and Cognition Center, Paris, France; [4]Institut Universitaire de France, Paris, France

**Abstract** Motor variability is a fundamental feature of developing systems allowing motor exploration and learning. In human infants, leg movements involve a small number of basic coordination patterns called locomotor primitives, but whether and when motor variability could emerge from these primitives remains unknown. Here we longitudinally followed 18 infants on 2–3 time points between birth (~4 days old) and walking onset (~14 months old) and recorded the activity of their leg muscles during locomotor or rhythmic movements. Using unsupervised machine learning, we show that the structure of trial-to-trial variability changes during early development. In the neonatal period, infants own a minimal number of motor primitives but generate a maximal motor variability across trials thanks to variable activations of these primitives. A few months later, toddlers generate significantly less variability despite the existence of more primitives due to more regularity within their activation. These results suggest that human neonates initiate motor exploration as soon as birth by variably activating a few basic locomotor primitives that later fraction and become more consistently activated by the motor system.

**\*For correspondence:**
elodie.hinnekens@gmail.com

[†]These authors contributed equally to this work

**Competing interest:** The authors declare that no competing interests exist.

## Editor's evaluation

This important work on locomotor development takes a longitudinal approach to show that the number of basic locomotor 'primitives' in infant stepping increases from newborn to walking onset, while the variability in their activation decreases. It presents convincing data from the modelling of EMG and kinematic data, which should be of interest to physiologists and psychologists interested in motor skills and development.

## Introduction

Variability arises at several levels of the motor system during early locomotor development. Firstly, as soon as birth, infants are able to perform a wide range of behaviors involving flexion and extension cycles of the lower limbs, such as stepping, kicking, swimming, or crawling (*Forma et al., 2019*; *McGraw, 1941*; *McGraw, 1939*; *Sylos-Labini et al., 2020*; *Thelen et al., 1983b*; *Thelen and Fisher, 1982*).Secondly, a given behavior can be realized with numerous coordination modes. For example, neonatal stepping can involve alternated steps, parallel steps, serial steps, or single steps (*Siekerman et al., 2015*). Similarly, toddlers can follow curved paths when walking or generate a variety of coordination patterns on the fly when cruising over varying distances (*Ossmy and Adolph, 2020*). Thirdly, a given coordination mode can be realized by different combinations of muscles. For example, infants demonstrate a high variability of muscle activations throughout their first year of life when stepping

**eLife digest** Human babies start to walk on their own when they are about one year old, but before that, they can move their legs to produce movements called 'stepping', where they take steps when held over a surface; and kicking, where they kick in the air when lying on their backs. These two behaviors are known as 'locomotor precursors' and can be observed from birth.

Previous studies suggest that infants produce these movements by activating a small number of motor primitives, different modules in the nervous system – each activating a combination of muscles to produce a movement. However, babies and toddlers exhibit a lot of variability when they move, which is a hallmark of typical development that furthers exploring and learning. So far, it has been unclear whether such differences arise as soon as babies are born and if so, how a small number of motor primitives could result in this variability.

Hinnekens et al. hypothesized that the great variety of movements in infants can be generated from a small set of motor primitives, when several cycles of flexing and extending the legs are considered. To test their hypothesis, the researchers first needed to establish how and when infants generate this variability of movement. To do so, they used electromyography to record the leg muscle activity of 18 babies during either movement resulting in a body displacement (locomotor movement) or rhythmic movement. These measurements were taken at either two or three timepoints between birth and the onset of walking.

Next, the scientists used a state-of-the-art machine learning approach to model the neural basis underlying these recordings, which showed that newborns generate a lot of movement variability, but they do so by activating a small number of motor primitives, which they can combine in different ways. Hinnekens et al. also show that as babies get older, the number of motor primitives increases while the variety of movements decreases due to a more steady activation of each motor primitive.

Cerebral plasticity is maximal during the first year of life, and infants can regularly learn new motor skills, each leading to the ability to perform more movements. Motor variability is believed to play an important role in this learning process and is known to be decreased in atypical development. As such, examining motor variability may be a promising tool to identify neurodevelopmental delays at younger ages.

or kicking, even when producing only alternated leg movements (*Sylos-Labini et al., 2020*; *Teulier et al., 2012*). This multilevel variability can arise in numerous environmental contexts and is associated with the development of multiple components, like the growth of musculoskeletal structures, the myelination of neural circuits, or the motivational goal to move, leading infants to learn new skills with their own developmental time scale (*Adolph et al., 2018*).

The third type of variability, corresponding to the ability of the human body to produce a given movement in various ways, is permitted by the existence of numerous nerves and muscles that can control a given joint, which is often referred to as motor control redundancy (*Bernstein, 1967*). In adulthood, the central nervous system (CNS) seems to simplify the coordination of these numerous degrees of freedom (DOFs) via a small number of encoded primitives, also called motor modules or muscle synergies (*Bizzi et al., 1991*; *d'Avella et al., 2003*; *Tresch et al., 1999*). A primitive is a neural structure that is stored within the CNS at a spinal level and that autonomously produces a coordinated pattern of behavior (i.e. involving several muscles) when recruited from higher centers (*Bizzi et al., 2008*). In adult organisms, primitives seem to be encoded within the spinal cord and the brainstem (*Bizzi et al., 1991*; *Hart and Giszter, 2010*; *Mussa-Ivaldi et al., 1994*; *Roh et al., 2011*) and activated by the motor cortex (*Drew et al., 2008*; *Overduin et al., 2015*; *Overduin et al., 2012*) as well as regulated by sensory feedback (*Cheung et al., 2005*). In humans, the physical location of such primitives remains unknown, but computational modeling from electromyographic (EMG) data also suggests the existence of a modular command (*Berger et al., 2013*; *Ivanenko et al., 2004*). Two types of modules are described: a spatial module is a group of muscles that are activated together with relative weights, while a temporal module is a waveform that describes the activation of a spatial module across time (*Delis et al., 2014*). In adults walking, the EMG activity of numerous muscles of the lower limb can be efficiently reproduced by 4–5 spatial and temporal modules (*Dominici et al., 2011*; *Hinnekens et al., 2020*; *Lacquaniti et al., 2012*; *Neptune et al., 2009*; *Figure 1*).

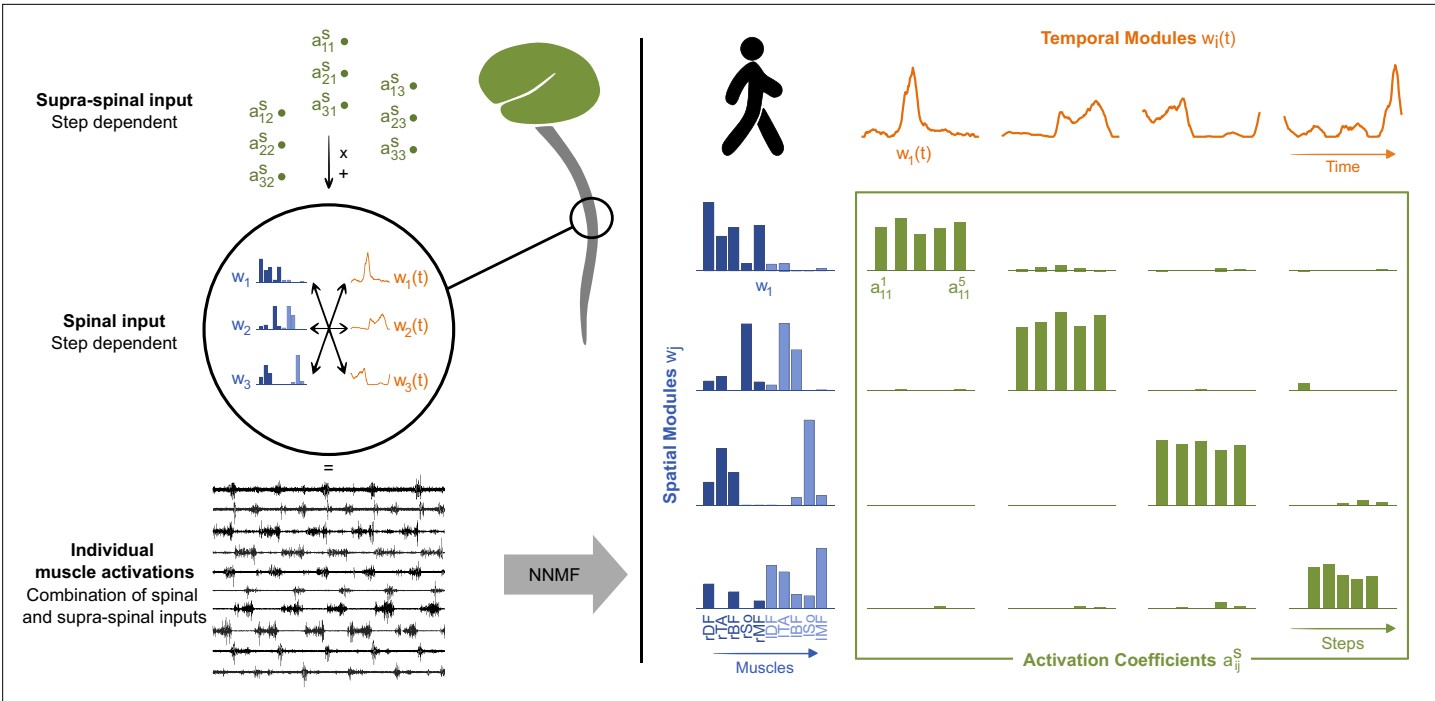

**Figure 1.** Theory of modularity and modular organization of adult walking. Left: the theory of modularity postulates that individual muscle activations result from the combination of basic spinal structures called locomotor primitives, which are of two types: spatial (blue) and temporal (orange) modules. According to the space-by-time model that is used here, the brain activates those modules through a supraspinal input (green) that specifies which amplitude of activation has to be allocated to each possible pair of spatial and temporal modules. In humans, non-negative matrix factorization (NNMF) is used to identify the underlying motor primitives and their activation coefficients from electromyographic (EMG) data. Right: illustration of NNMF applied to five right steps of walking in a human adult. EMG patterns can be decomposed into four spatial modules (blue) and four temporal modules (orange). Muscles from both sides can be allocated to a same spatial module to form bilateral modules. Within each spatial module, weightings are plotted for muscles $m_1$ to $m_{10}$ in the following order: rectus femoris, tibialis anterior, biceps femoris, soleus, and gluteus medius (right muscles in dark blue followed by left muscles in light blue). Activation coefficients (green) represent the level of activation of each possible pair of spatial and temporal modules during five steps. Two features are typical of adults' modular organization: the stability of activation (activations coefficients remain stable during the five steps) and the selectivity of activation (one spatial module is always activated with only one temporal module and vice versa).

The development of this fine-tuned modular organization has been investigated from birth on, and several neonatal behaviors have been found to already rely on a low-dimensional modular organization. In particular, stepping and kicking are two neonatal behaviors that can involve alternate flexion and extension cycles of the lower limb, stepping being elicited by a pediatrician when the infant is held in an erected position while kicking is a natural behavior performed in supine position. Those behaviors were found to each involve modules that present similarities with mature modules, making them both distinct locomotor precursors (*Dominici et al., 2011*; *Sylos-Labini et al., 2020*). For example, neonatal stepping is based on two modules while walking in toddlers is based on four, suggesting that the motor repertoire of newborns is restricted (*Dominici et al., 2011*). This is coherent with the fact that innate behaviors are described as stereotyped (*Jeng et al., 2002*; *Spencer and Thelen, 2000*; *Thelen et al., 1981*), with strong coupling among joints (*Fetters et al., 2004*; *Jeng et al., 2002*; *Thelen et al., 1981*) and among agonist/antagonist muscles (*Teulier et al., 2012*), while typically developing infants will develop the ability to dissociate their degrees of freedom toward a richer repertoire (*Fetters et al., 2004*). However, the existence of a low-dimensional modular organization in newborns was established on single-step or averaged data (*Dominici et al., 2011*; *Sylos-Labini et al., 2020*), while the muscular activity that underlies neonatal movements is known to be highly variable, even for a given coordination mode like, for example, across alternated leg movements (*Teulier et al., 2012*). This intra-individual variability is believed to be a key feature of typical motor development allowing learning (*Dhawale et al., 2017*; *Hadders-Algra, 2018*). According to animal studies, variability could even be centrally regulated for the purpose of motor exploration (*Kao et al., 2008*; *Mandelblat-Cerf et al., 2009*).

If modularity and variability seem antagonistic at first glance, the question of whether those two features are compatible or incompatible needs to be addressed to better understand motor development. On one hand, a low-dimensional modular system inherently limits motor exploration (*Cohn et al., 2018*; *Valero-Cuevas, 2009*). In this vein, the high variability of EMG data during the first year of life opened discussion about the existence itself of motor primitives (*Teulier et al., 2012*). On the other hand, data from animals suggest that such variability could be generated within a modular system during development. A variable output was indeed observed after applying different stimuli to the neonatal spinal cord of rodents (*Kiehn and Kjaerulff, 1996*; *Klein et al., 2010*), which is also believed to store motor primitives (*Blumberg et al., 2013*; *Dominici et al., 2011*). In young songbirds, a specialized cortical area has even been found to be responsible for inserting variability into the temporal structure of vocalization to facilitate learning in early development, resulting in a highly variable output that becomes structured when inhibiting the area (*Aronov et al., 2011*; *Kao et al., 2008*). As those data suggest that is possible to produce variability by modulating the activation of basic inputs, such organization might shape the development of the motor system in human infants.

In human infants, investigations of the motor system are more limited and EMG recordings are the closest signals to the neural output that can be recorded while moving. Yet, if modularity and variability do coexist within the neural command, we should be able to separate the contribution of motor primitives from the variability of EMG signals and observe their cross-evolution during development. To test this prediction, we longitudinally followed 18 human infants and recorded the EMG activity of 10 lower-limb muscles on 2–3 time points between birth and walking onset, during stepping, kicking, or walking (*Figure 2*). Using a state-of-the-art unsupervised machine learning approach, we were able to model the underlying command by decomposing the EMG data of numerous muscles into step-invariant basic muscle patterns (which represent the motor primitives at each age) and into step-variable activation coefficients (which theoretically represent the variable descending command that modulates the activation of the motor primitives, at least in adults) (*d'Avella et al., 2003*; *Delis et al., 2014*; *Figure 1*). We describe the evolution of both motor variability and motor modularity from birth to independent walking and provide evidence that the human motor system could theoretically initiate its exploration by variably activating a few temporary basic structured patterns.

## Results

Eighteen infants were tested longitudinally on 2–3 time points between birth (~4 d) and walking onset (~14 mo). The time points were either around birth, around 3 mo, or around walking onset (individual characteristics and precise time points are reported in Table 2). Around birth and 3 months old, we observed the stepping behavior and/or the kicking behavior, while at walking onset we only recorded independent walking. In each behavior and at each age, infant movements were recorded using surface EMG on 10 bilateral lower-limb muscles and two 2D video cameras. Based on the resulting films, trained coders selected alternated cycles of flexion and extension of the lower limbs, which allowed us to study the same movement regardless of the behavior that could be produced by the infant at each age and focus only on the generation of trial-to-trial variability for this given movement. Data from a given baby were considered analyzable when we had recorded clean surface EMG signals of the 10 lower-limb muscles during at least five alternated cycles of flexion and extension, both at birth and 3 months old and through the same behavior (stepping or kicking). Those cycles were not necessarily consecutive, but to be selected a given cycle had to be at least preceded by an extension and succeeded by a flexion. In total, 586 cycles of flexion and extension were included into the analysis. When more than five cycles were available, we proceeded by analyzing random combinations of five cycles among the available ones and averaging the results afterward, so that the variability would always be calculated on a same number of cycles. For each behavior, we computed the variability of the motor output (index of EMG variability [IEV]) and used non-negative matrix factorization (NNMF) to identify the underlying motor primitives and their activation parameters. We computed a goodness-of-fit criterion to establish whether the cycle-to-cycle variability of five cycles of flexion and extension of the lower limbs could be produced through various combinations of those motor primitives. We compared this goodness-of-fit criterion across ages and computed other indexes in order to characterize (1) how variably were those motor primitives activated and (2) how selective were those primitives (i.e. if they controlled numerous muscles at a time or a few muscles). *Table 1* summarizes the role of each of the main variables. Details are available in the 'Materials and methods' section.

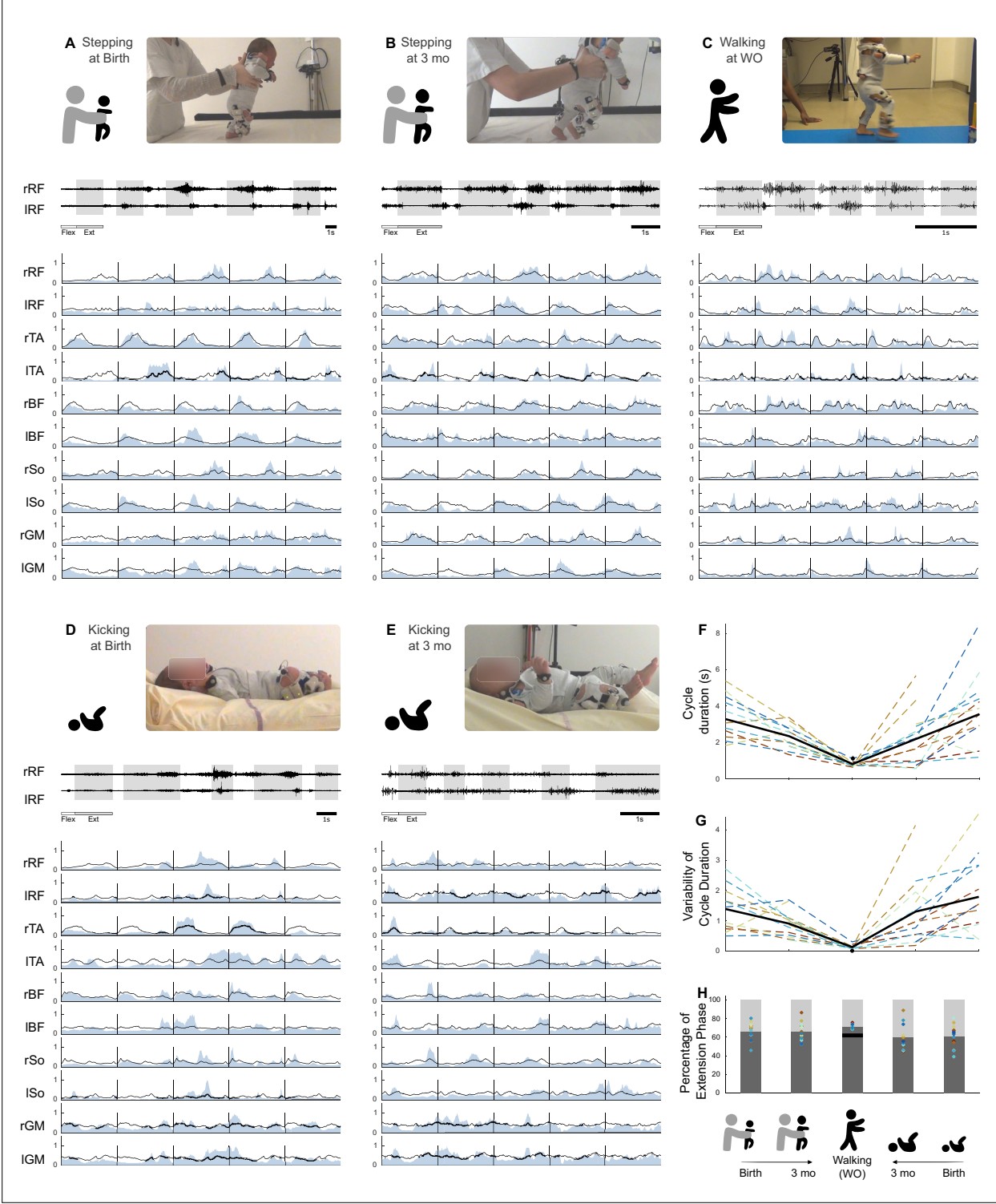

**Figure 2.** Development of basic electromyographic (EMG) and kinematic parameters. (**A–E**) Example of EMG data for each age and behavior in one infant. A set of five cycles of flexion and extension is presented for each age and behavior. High-pass-filtered data are shown for two muscles (extension phases appear on a gray background). The 10 muscles are then pictured as completely preprocessed (i.e. filtered and normalized in amplitude and time, blue envelope). The black line is the averaged signal across the five pictured cycles. The scale of 1 s is displayed at the bottom of each figure. RF, rectus femoris; TA, tibialis anterior; BF, biceps femoris; So, soleus; GM, gluteus medius. (**F–I**) Evolution of several features starting from birth to walking onset for stepping or kicking. Individual data are shown in dotted line with the same color code as in *Figure 3*. Each point was computed as a mean score for each individual (see section 'Number of cycles included in the analysis'). The black bold line represents the averaged values across individuals. The black

*Figure 2 continued on next page*

*Figure 2 continued*

point (or trait in **F**) represents the adult landmark. (**F**) Cycle duration. (**G**). Kinematic variability (standard deviation of cycle duration divided by averaged cycle duration). (**H**) Proportion of flexion and extension phases.

The online version of this article includes the following source data for figure 2:

**Source data 1.** Individual data regarding basic electromyographic (EMG) and kinematic parameters (corresponding to *Figure 2F–H*).

## Kinematic parameters and EMG signals reveal maximal motor variability during the neonatal period

We started by characterizing basic kinematic and EMG parameters at each age and in each behavior. Wilcoxon tests were performed among kinematic parameters (cycle duration and its variability, proportion of extension/flexion phases) to assess basic differences. The cycle duration was different across behaviors with a decrease from stepping at birth to stepping at 3 mo (p=0.01) and to walking in toddlers (p<0.001) as well as a decrease from kicking at birth to kicking at 3 mo (p=0.003) and to walking in toddlers (p=0.009, *Figure 2F*). The proportion of phases within a cycle was slightly different across ages (*Figure 2H*, *Supplementary file 1a*).

The kinematic variability was assessed by the variability of cycle duration (*Figure 2G*). This variability significantly decreased for stepping and kicking from 3 mo to walking onset in toddlers (p<0.001). Between birth and 3 months old, this variability seems to have begun to decrease (p=0.021 for kicking, and trend of p=0.083 for stepping). The variability of EMG data was assessed by the IEV ( *Figure 3D*). This index significantly decreased from 3 mo to walking onset in both stepping and kicking (p=0.003 and p=0.001 respectively). However, it significantly decreased between birth and 3 months old for stepping (p=0.005) and not for kicking in which the evolution seems to be different across individuals (p=0.519, *Figure 3D*).

## The number of motor primitives increases from birth to walking onset while variability decreases

A modular decomposition was applied to each dataset (for a given behavior, at a given age and for a given subject) thanks to NNMF. We found that several aspects of this decomposition were different depending on the age regarding both dimensionality (i.e. number of primitives) and variability of activations (*Figures 3 and 4*).

To study dimensionality, we considered two approaches based on the variance accounted for (VAF) that is the index that indicates the quality of the modelling. The first approach identified the number of motor primitives (i.e. spatial and temporal modules) that are needed to reach a predetermined

**Table 1.** Summary of the role of the main variables of the study.

| Short name | Role |
| --- | --- |
| Index of EMG variability (IEV) | Represents the cycle-to-cycle variability of EMG data across five alternated cycles of flexion and extension of the lower limb. |
| Variance accounted for (VAF) | Represents the goodness of fit of the model of modularity for a given number of modules. When the VAF for a fixed number of four spatial and temporal modules is computed, it quantifies how well experimental data can be modeled as originating from four modules. |
| Number of modules | Represents the smallest number of invariant spatial and temporal elements in which the EMG signals can be factorized (chosen as the smallest number allowing to reach a VAF > 0.75). |
| Index of recruitment variability (IRV) | Represents the extent to which spatial and temporal modules are steadily (lower value) or variably activated across cycles to produce the EMG outputs (higher value). |
| Index of recruitment selectivity (IRS) | Represents the extent to which spatial modules can be activated with different temporal modules (lower values) or exclusively activated with a given temporal module (higher value). |
| Selectivity of muscular activations index (SMAI) | Represents the extent to which spatial modules each control numerous muscles at a time (lower value) or a few muscles at a time (higher value). |
| Selectivity of temporal activations index (STAI) | Represents the extent to which temporal modules each control muscles during a long time (lower value) or during a shorter peak of time (higher value). |

EMG, electromyography.

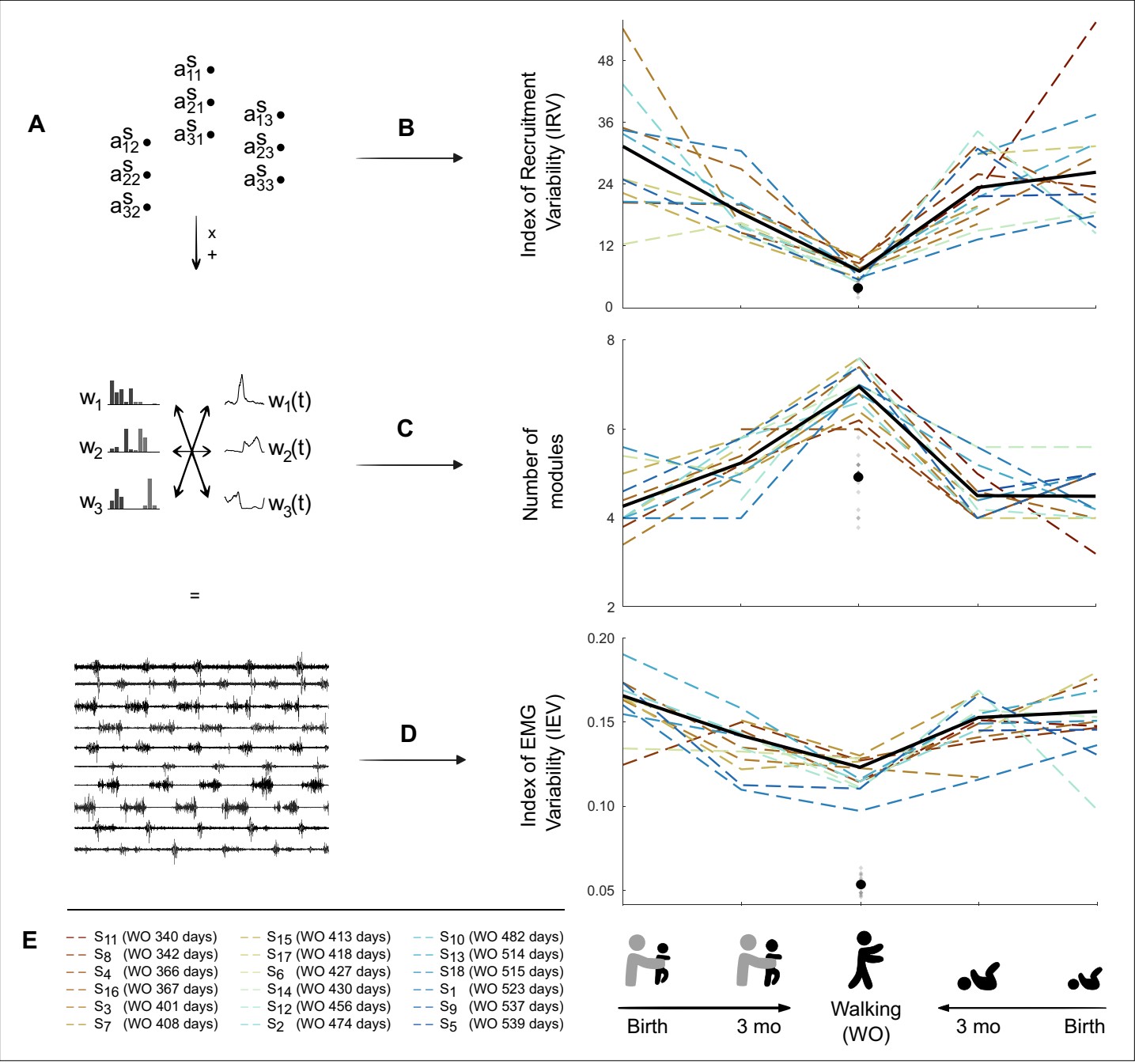

**Figure 3.** Decrease in variability between birth and walking onset associated with modifications of the underlying set of motor primitives. (**A**) Computational elements contributing to the electromyography (EMG) and their trial-to-trial variability (from top do down: activation coefficients, spatial and temporal modules, and muscle outputs). (**B–D**). Graphs (**B–D**) show how changes within the upper levels can explain the resulting motor variability during infant locomotor development. Individual data are represented as dotted lines. Each point was computed as a mean score for each individual (see section 'Number of cycles included in the analysis'). The black bold line represents the averaged values across individuals. The black point indicates the adult landmark, and the gray diamonds indicate individual values from 20 adults (**Supplementary file 1c**). (**B**) Variability of module activations, assessed by the index of recruitment variability (IRV). IRV represents the variability of the input that specifies which amplitude of activation has to be allocated to each possible pair of spatial and temporal modules. This index decreases from birth to walking onset considering stepping or kicking as neonatal behavior. (**C**) Number of spatial and temporal modules, which increases from birth to walking onset considering stepping or kicking as neonatal behavior (**D**) Index of EMG variability (IEV, same as in **Figure 2I**). This index decreases from birth to walking onset, considering stepping or kicking as neonatal behavior. (**E**) Figure legend. Each individual is represented by a color throughout the article. To take into account the variability of walking onset in our representations, colors of each individual are sorted according to their age of walking onset.

The online version of this article includes the following source data for figure 3:

*Figure 3 continued on next page*

*Figure 3 continued*

**Source data 1.** Individual data regarding electromyographic (EMG) output and modeling of the modular organization from birth to walking onset (corresponding to *Figure 3B–D*).

VAF threshold. This threshold was established to 0.75 according to *Hinnekens et al., 2020*. This approach allowed us to determine the number of modules of each individual, which showed that the number of modules was higher at walking onset than at birth and 3 mo (the number of modules was on averaged 4.3 ± 0.7 for stepping and 4.6 ± 0.6 for kicking at birth, 5.2 ± 0.6 for stepping, and 4.5 ± 0.6 for kicking at 3 months old, and 7 ± 0.6 for walking at walking onset, *Figure 3C*). The second approach set the number of modules to four as in standard adult walking and relied on the analysis of the resulting VAF. This approach assessed dimensionality of the underlying modular system just like the first one but directly tested the hypothesis that four spatial and temporal modules are sufficient to adequately represent the given EMG signals across cycles. By relying on real numbers instead of integers, this second approach is useful because it is more suited to perform statistical analyses. It confirmed that a low-dimensional model fitted better at birth than at walking onset (*Supplementary file 1b*). We observed a significant VAF decrease between stepping at birth and walking (p=0.002) and between stepping at 3 mo and walking (p<0.001), with the same effects for kicking (p<0.001). Between birth and 3 months old, the VAF value significantly decreased in stepping (p=0.019) but not in kicking (p=0.850) for which the evolution was different across individuals (*Figure 3C*). To sum up, the modular organization was more complex in toddlers than in infants, with a decrease in the VAF with age, indicating that more and more modules were needed to equivalently reconstruct the EMG patterns, as illustrated by *Figures 3B and 4*.

## Motor primitives are recruited with maximal variability and low selectivity during the neonatal period

After having analyzed the dimensionality of the signals, we wanted to explain how the IEV (EMG variability) could be higher in infants while their dimensionality was lower. Thus, we focused on the variability of activations of motor primitives. The index of recruitment variability (IRV), which represents the extent to which spatial and temporal modules are variably activated across steps, significantly decreased in toddlers in comparison to infants (*Figure 3A*), indicating that module recruitment was less and less variable starting from either stepping or kicking from birth to walking (respectively p=0.002 and p<0.001) and from 3 mo to walking (p<0.001). Here again, the value significantly decreased between birth and 3 months old for stepping (p=0.001) but not for kicking (p=0.424). To check that the effects were not due to differences in the number of modules, we performed the same computations on values obtained by systematically extracting four spatial and temporal modules and found the same effects (*Supplementary file 1a*). This shows that, even with the same number of modules, toddlers, almost like adults recruit modules in a more systematic way across cycles than infants (see *Figure 3—source data 1* for individual data). Finally, we repeated the analysis while allowing the modules to vary for each cycle, similarly to what was done in *Cheung et al., 2020a*, and still found the same effect on the IRV (see *Figure 4—figure supplements 1 and 2*).

The index of recruitment selectivity (IRS, which represents the extent to which a spatial module is activated with a single temporal module and vice versa), tended to increase with age, ranging from 0.395 on average in newborn stepping or kicking to 0.44 in toddlers walking (*Supplementary file 1b*). This index was always far below the adult value at every age, which is on average 0.62 (*Supplementary file 1c*), suggesting a low selectivity in the recruitment of spatial and temporal modules during development. Indeed, a spatial module could be activated along with several temporal modules and vice versa depending on the cycle (*Figure 4*). We also repeated this computation after having extracted four spatial and temporal modules from each dataset and found the same results.

## Motor primitives evolve between birth and walking onset toward gathering less muscles at a time

In order to identify whether motor primitives would have been preserved across ages, we applied the best matching pairs method (*Cheung et al., 2005*) to our data and checked for similitudes between modules. As we could not find high similitudes, we noticed that modules seemed to be less and less

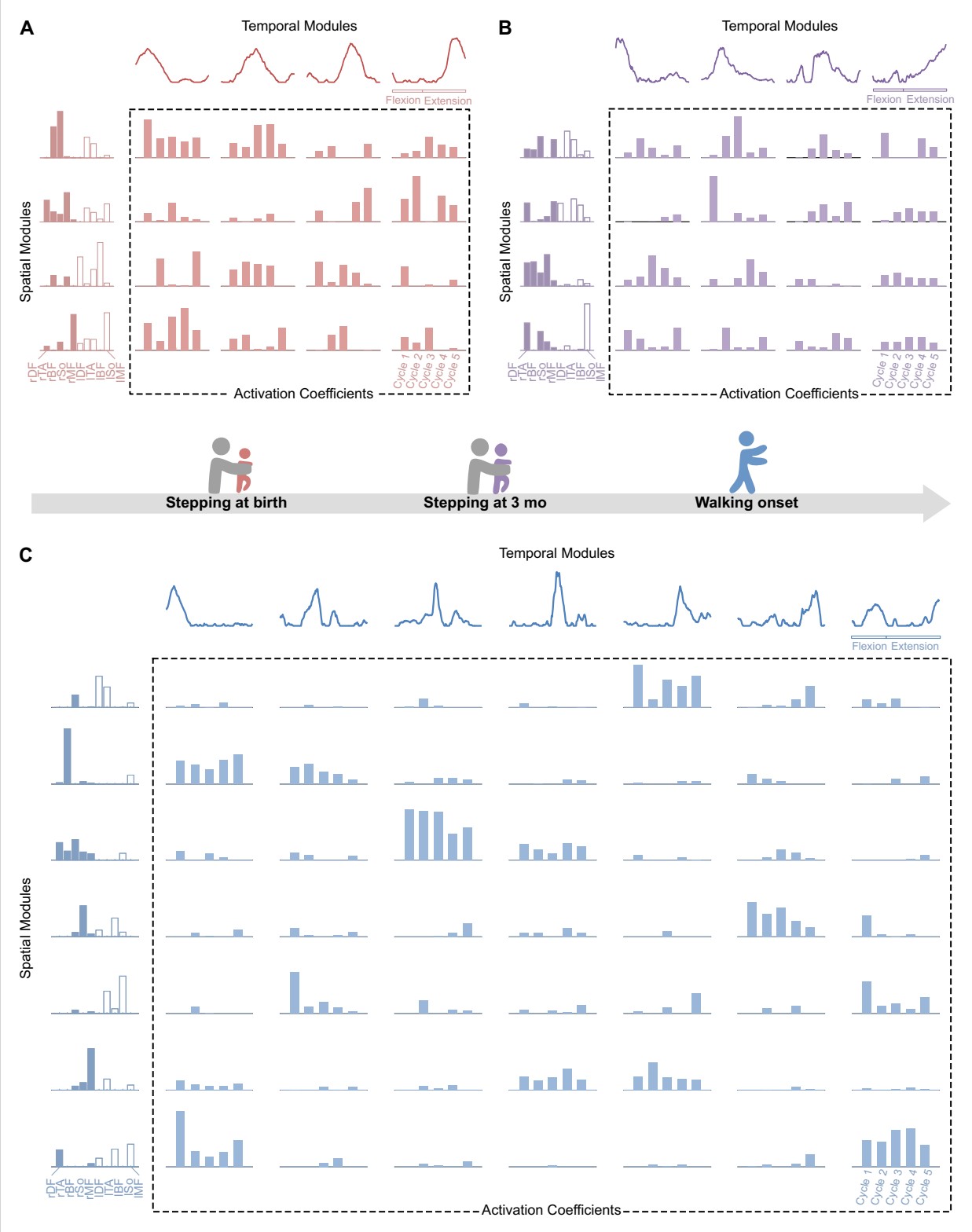

**Figure 4.** Modular organization at each age in a representative individual. At each age, electromyographic (EMG) patterns can be decomposed into spatial modules and temporal modules (orange). Within each spatial module, weightings are plotted for muscles $m_1$ to $m_{10}$ in the following order: rectus femoris, tibialis anterior, biceps femoris, soleus, and gluteus medius (right muscles in dark colors followed by left muscles in light colors). Activation coefficients (at the crossing between each spatial and temporal modules) represent the level of activation of each possible pair of spatial and temporal modules during five steps. (A-B) At birth (red, top left) and 3 mo (purple, top right), EMG activity of stepping can be decomposed into four spatial and

*Figure 4 continued on next page*

*Figure 4 continued*

four temporal modules. (**C**) At walking onset (blue, bottom), EMG activity needs to be decomposed into seven spatial and seven temporal modules to get the same quality of modeling than at birth and 3 mo with less modules. Activation coefficients are highly variable at birth and 3 mo and less variable in toddlerhood, with some pairs that are nearly never activated across the five cycles. Note that toddler activations are still more variable than in adults (*Figure 1*).

The online version of this article includes the following figure supplement(s) for figure 4:

**Figure supplement 1.** In our analysis, we implicitly assumed that modules cannot vary from cycle to cycle (very short time scale) but that they can vary across development (long time scale).

**Figure supplement 2.** Using the approach presented in *Figure 4—figure supplement 1*, we computed the plasticity of spatial modules, temporal modules, and activation coefficients (i.e. the IRV, Index of Recruitment Variability) for the specific number of modules of each individual and age (**A–C**) and for a number of modules fixed to 4 (**D–F**).

complex with time, suggesting a more individual muscle control (i.e. spatial modules gathered the activation of less and less muscles, and temporal modules represented tighter and tighter peaks of activation, *Figure 5A*). To quantify this phenomenon, we created two indexes: the selectivity of muscular activations indexes (SMAI) and the selectivity of temporal activation indexes (STAI) (see *Figure 5—source data 1* for individual data). The SMAI increased after 3 mo, indicating that muscle weightings were sparser among spatial modules in toddlers than in infants. In other words, spatial modules were mostly composed of fewer muscles in toddlers compared to infants (see *Figure 5A and B*). This increase occurred from stepping and kicking at birth to walking (p=0.002 and p<0.001, respectively) and from stepping or kicking at 3 mo to walking (p<0.001). No significant effect was found between birth and 3 mo for stepping or kicking even if a trend appeared for stepping (p=0.054).

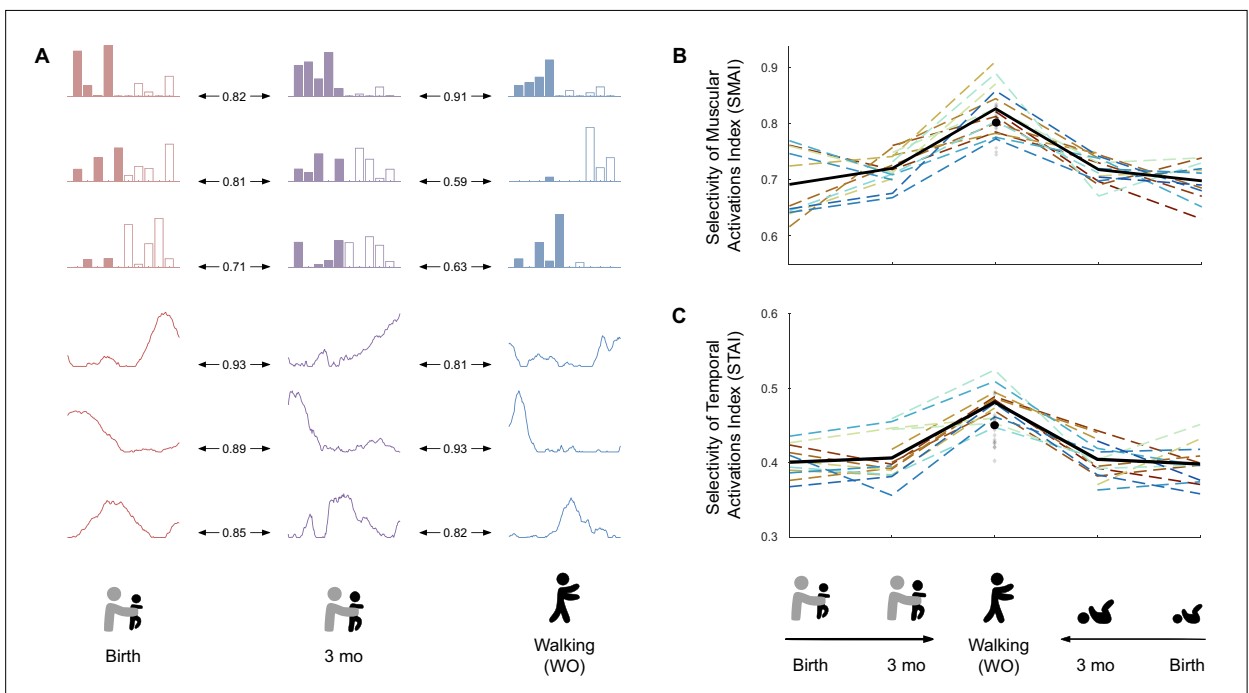

**Figure 5.** Development of modules' structure from birth to walking onset. (**A**) Similarity of modules between ages in a given individual according to the best matching pair method. (**B**) Selectivity of muscular activations index (SMAI). (**C**) Selectivity of temporal activations index (STAI). Both indexes increase between birth and walking onset considering stepping or kicking as neonatal behavior. Individual data are shown in dotted line with the same color code as in *Figure 3*. Each point is computed as a mean score for each individual (see section 'Number of cycles included in the analysis'). The bold line represents the averaged values across individuals. The black point indicates the adult landmark, and the gray diamonds indicate individual values from 20 adults (*Supplementary file 1c*).

The online version of this article includes the following source data for figure 5:

**Source data 1.** Individual data regarding modules' structure from birth to walking onset (corresponding to *Figure 5B and C*).

Analyses were repeated on modular decompositions coming from the systematic extraction of four spatial and temporal modules and gave the same results (*Supplementary file 1a*).

The STAI also increased from birth and 3 mo to walking. This indicates that temporal modules had more tightened peaks of activation in toddlers than in infants (*Figure 5A and C*). The value for toddlers was also above the adult landmark (*Figure 5C*). For both stepping and kicking, the increase occurred between birth and walking (p=0.004 and p<0.001, respectively) and between 3 mo and walking (p<0.001). Analyses were repeated following the extraction of four spatial and temporal modules and gave the same results (*Supplementary file 1a*).

## Discussion

The objective of this study was to investigate whether and when trial-to-trial variability could emerge from the modular system of developing human infants. We found that the EMG variability of lower-limb muscles was maximal in the neonatal period and decreased at walking onset. This decrease in trial-to-trial variability was associated with an increase in the number of motor primitives. In the neonatal period the maximal EMG variability could be explained by variable activations and combinations of a small number of motor primitives, suggesting that the motor system can generate trial-to-trial variability at birth albeit relying on few primitives. In contrast at walking onset, more primitives were needed to explain EMG patterns of alternated leg movements, and these primitives were activated more consistently across trials. Furthermore, primitives were more selective than in infanthood, suggesting that toddlers developed the capacity to control less muscles at a time. Together, these results provide evidence that the motor system is flexible as soon as birth despite the existence of basic patterns of coordination and suggest that neonatal motor primitives are plastic structures that fraction in early life to become more selective and more steadily activated. Below we discuss these findings in relation with the current literature about motor exploration and the maturation of the motor system through development.

This study highlights that EMG signals of alternated leg movements in human infants are highly variable as soon as birth, while neonatal behavior is often described as rather stereotyped due to the high prevalence of cocontractions (*Dominici et al., 2011*; *Jeng et al., 2002*; *Spencer and Thelen, 2000*; *Thelen et al., 1981*). Moreover, the signals analyzed here showed high variability while they were screened and selected among a lot of cycles that could be parallel or unilateral rather than alternate, which demonstrates even more flexibility than what we analyzed (*Siekerman et al., 2015*; *Thelen and Fisher, 1983a*; *Thelen et al., 1981*). If we assume that trial-to-trial variability is a marker of motor exploration (*Dhawale et al., 2017*; *Kao et al., 2008*), our results would suggest that exploring the field of possible movements is possible as soon as birth despite the difficulty in uncoupling the different muscles and joints of lower limbs (*Fetters et al., 2004*; *Teulier et al., 2012*; *Vaal et al., 2000*). Such idea is coherent with the fact that motor learning abilities are in place very early in development and even before birth (*Robinson, 2015*; *Robinson and Kleven, 2005*).

Our results revealed that the most variable EMG output was seen at the time where the fewest number of primitives was found (i.e. in the neonatal period). This confirms and extends previous analyses relying on cross-sectional data showing that the motor system involves a few modules in rat and human newborns when producing alternated leg movements (*Dominici et al., 2011*; *Sylos-Labini et al., 2020*) and is in coherent with recent results showing that apparently noisy data could stem a low-dimensional organization in human infants (*Sylos-Labini et al., 2022*). It also shows that trial-to-trial variability in humans can be generated through a structured modular system similarly to newborn rats (*Blumberg et al., 2013*), which seems to be specific to infanthood (as opposed to toddlerhood) according to our results. Although the neural interpretation is limited here, motor primitives were originally identified as spinal structures in animals (*Bizzi et al., 1991*) and are still hypothesized to be stored at a spinal level and activated through supraspinal inputs in adults (*Bizzi and Cheung, 2013*; *Delis et al., 2014*; *Overduin et al., 2015*; *Overduin et al., 2012*; *Roh et al., 2011*). As such our results are reminiscent with neurophysiological studies in rodents that demonstrate that different motor patterns can emerge by applying different pharmacological or electrical stimuli to the neonatal spinal cord (*Kiehn and Kjaerulff, 1996*; *Klein et al., 2010*) and that the neonatal motor system is capable of flexibility via the variable activations of a few basic structures (*Blumberg et al., 2013*). Such variable activations could be due to the instability of the command and be incidental, but it could also be genuinely purposeful: in young birds learning to sing, the output appears to be highly variable, but

becomes even more structured than the adult one after having inhibited some specific areas of the brain, proving that such areas were responsible for generating variability and facilitating exploration (*Aronov et al., 2011*). With growth, adaptive inhibitory mechanisms settle in a way that does not compromise the capacity for future plasticity (*Garst-Orozco et al., 2014*). As similar neurophysiological investigations cannot be conducted in human infants, discriminating among purposeful and incidental variability remains challenging. However, the present results demonstrate that the motor output of neonates can also be decomposed into highly structured patterns and their temporary variable activations, which shed new light about the potential existence of similar mechanisms. As we observed such structure within alternated leg movements, other studies are needed to explore the extent of these results to other early behaviors or coordination modes, as well as their link with the ability to move in various environmental contexts.

Strikingly, we observe the reverse phenomenon in toddlerhood: as infants grew to become toddlers, they seem to have widened their motor repertoire along with a stabilization of their command to produce alternate leg movements. Indeed, more modules were needed to explain the EMG signals than in infanthood (*Figure 3B*), similarly to what was found in *Dominici et al., 2011*. The same trend was observed when considering stepping or kicking, which were both shown to be neonatal loco-motor precursors (*Sylos-Labini et al., 2020*). The increase in the number of modules along the first year of life was associated with the changes within the shape of modules themselves. They became more selective at walking onset (*Figure 5*), suggesting a more individual muscle control. This is in accordance with previous results reporting fractionation of primitives and decrease of duration of temporal modules (*Sylos-Labini et al., 2020*) as well as consistent with the known decrease in cocon-tractions across the first year of life (*Teulier et al., 2012*). This is also strongly reminiscent with recent findings regarding the development of running, which starts with a fractionation of motor primitives between childhood and adulthood, allowing those primitives to be merged again later with training (*Cheung et al., 2020a*). On a different scale of analysis that was recently allowed by the use of high-density EMG in human neonates, it was also established that the synchronization across motoneurons of a given muscle was more important in neonates than in adults, which suggests a reshaping of central pattern generators during development (*Del Vecchio et al., 2020*). Assuming that modules are stored at a spinal level, such a reshaping within modules themselves is likely to be made early in infancy. Spinal circuits are indeed particularly plastic during early development thanks to the high activity of neurophysiological processes that can lead to changes within neurons' excitability through practice (*Brumley et al., 2015*; *Vinay et al., 2002*). As neonates are able to set up several behaviors that involve flexion and extension cycles of lower limbs, such as stepping and kicking but also air-stepping, crawling, or swimming (*Barbu-Roth et al., 2014*; *Forma et al., 2019*; *Forma et al., 2018*; *McGraw, 1941*; *McGraw, 1939*), they could already benefit from a lot of opportunities to explore and practice in order to shape the underlying circuits. The particular plasticity of motor primitives at this point is most likely allowed by the absence of stable synaptic connections within pathways, synaptogenesis being particularly active in the neonatal period (*An et al., 2012*; *de Graaf-Peters and Hadders-Algra, 2006*).

The cross-development between trial-to-trial variability of activations and the number of primitives suggests that the motor system is never designed to produce a maximal amount of variability. One could indeed picture a scenario where a maximal number of primitives would have been associated with maximally variable activations of these modules to maximize variability and exploration (or even no primitives at all to allow free muscle activations). However, constraining the space of possible options might be crucial for motor learning (*Bernstein, 1967*; *Dhawale et al., 2017*), which also relates to the exploration–exploitation tradeoff in reinforcement learning (*Sutton and Barto, 1998*). Humanoid robots indeed learn and build motor synergies more easily when starting to explore with a limited number of available degrees of freedom such as dynamic movement primitives (*Lapeyre et al., 2011*; *Lungarella and Berthouze, 2002*; *Schaal, 2006*). The developing system might have to always remain in an ideal compromise of constraint and flexibility, the difficulty of uncoupling lower-limb joints at the beginning of life (*Fetters et al., 2004*; *Vaal et al., 2000*) being an example of such helping constraint in humans. It was suggested that this rigid coupling that reduces the effective number of DOF allows to move with less need of processing capacities and without interferences of uncoordinated outputs (*Piek, 2002*). If this constraint might have a maturational origin, many factors could provide the same type of DOF reduction early in development (*Newell et al., 1989*), such as

environmental ones (e.g. the space within the womb environment or the gravity at birth) and factors related to the task (e.g. kicking might allow to focus on lower limbs only while walking also requires balance). Here we propose that the benefit of developmental motor primitives might be to yield an ideal space of possibilities that allow to efficiently explore several motor solutions via trial-to-trial variations of the activations of those temporary primitives. This might be critical to shape motor circuits, as suggested by the correlation between the lack of motor variability following early cerebral lesions and poor developmental outcomes in humans (*Einspieler and Prechtl, 2005*). In this context, the approach proposed here could contribute to the quantitative study of motor variability in atypical development (*Hadders-Algra, 2008*) and characterize the extent to which motor exploration can be elicited in developing early rehabilitation protocols that are based on rhythmic behaviors (*Angulo-Barroso et al., 2013*; *Campbell et al., 2012*; *Kolobe and Fagg, 2019*; *Sargent et al., 2020*; *Teulier et al., 2009*).

When working with EMG to identify hypothetical motor primitives, several factors can limit interpretations such as EMG processing, cross-talk, or arbitrary choices made during EMG factorization. It is indeed worth noticing that the current study reports different values than previous developmental studies regarding the absolute number of modules (*Dominici et al., 2011*; *Sylos-Labini et al., 2020*). However, there is currently no consensus regarding the selection of the number of modules that depends on arbitrary criteria such as the VAF threshold. We can also note that *Sylos-Labini et al., 2020* reported more modules in kicking than in stepping at birth in contrast to what is reported here; however, strict comparisons are limited on this point since we only studied supine kicking, whereas these authors considered kicking both supine and vertically held. Also, to deal with the lack of consensus about EMG processing, we reproduced our analysis with other choices regarding filtering, amplitude normalization, and time normalization, and verified that similar effects and trends emerged. This verification confirmed that our results were robust to reasonable processing choices. Cross-talk, which refers to the possibility of recording several muscles with one electrode, is particularly challenging in developmental studies since growth implies a modification of the distance across muscles. To deal with this issue, we purposely chose muscles from different body regions (the 10 muscles are distributed from the hips to the shank and over the two lower limbs). For agonist and antagonist muscles of the same body regions, we reproduced the analysis that was proposed by *Dominici et al., 2011* ensuring that raw signals were not correlated (see 'Materials and methods'). Despite all precautions, the size of surface EMG electrodes remains a challenging issue of the field, and current development of high-density EMG might offer interesting perspectives for collecting more and more precise signals through surface EMG (*Del Vecchio et al., 2020*).

Overall, when compared with adult values (*Figure 3*, *Supplementary file 1c*), our results suggest an immaturity of the modular system before and around walking onset, which confirms that infancy should be an ideal period of plasticity to benefit from in therapy (*Ulrich, 2010*; *Morgan et al., 2021*). These are also coherent with the idea that skill onset during development in not an on-off switch and that maturity is not reached as soon as a skill becomes possible (*Adolph et al., 2018*). Interestingly, the longitudinal design of this study highlighted some inter-individual differences among the evolution of kicking during the first months of life, whereas the overall variability of the motor output decreased in stepping. For example, the IEV decreased between kicking at birth and kicking at 3 months old for a part of the cohort while it increased for the other part (*Figure 3D*). These inter-individual differences were observed for the main variables of the study, including the number of primitives and the variability of activation (*Supplementary file 1a*). As kicking is a spontaneous behavior, these differences might relate with the fact that it is differently practiced across individuals. Some infants might indeed store more modules than others due to different amount of practice, while stepping would evolve toward more stable activations of more modules in every infant, coherently with the fact that the two behaviors are distinct locomotor precursors (*Sylos-Labini et al., 2020*). However, the longitudinal follow-up also showed that the structure of variability evolved in the same direction for every individual between 3 months old and walking onset, with the development of more modules that were more steadily activated, regardless of the locomotor precursor that was recorded at 3 months old. Interestingly, this indicates that the modular system seems to be in a common state for every individual around walking onset, despite the important variability of age in our cohort (*Table 2*). This state of high dimensionality might last for a non-negligible period of time. Recently, studying the development of walking and running in children, the authors identified more primitives at 2 and 5 years old than in adults (*Bach*

**Table 2.** Summary of individual characteristics.

| Subject ID | Gender | Birth weight (kg) | Birth height t (cm) | Age (days) | | | | Precursor that could be observed (T for treadmill, O for overground) |
| --- | --- | --- | --- | --- | --- | --- | --- | --- |
| | | | | First visit | Second visit | Walking onset | Third visit | |
| 1 | F | 3.05 | 48.5 | 21 | 91 | 523 | 535 | Stepping (T) and kicking |
| 2 | M | 4.02 | 52 | 2 | 98 | 474 | 499 | Stepping (T) and kicking |
| 3 | F | 3.71 | 52 | 3 | 87 | 401 | 412 | Stepping (O) and kicking |
| 4 | M | 3.46 | 48 | 2 | 98 | 366 | 371 | Stepping (O) and kicking |
| 5 | F | 3.07 | 50 | 2 | 86 | 539 | | Kicking |
| 6 | M | 3.74 | 51 | 2 | 74 | 427 | 434 | Stepping (O) and kicking |
| 7 | F | 3.68 | 50 | | 86 | 408 | 431 | Stepping (T) and kicking |
| 8 | M | 2.99 | 50 | 2 | 79 | 342 | 365 | Stepping (O) and kicking |
| 9 | M | 2.95 | 48 | 2 | 79 | 537 | 555 | Stepping (T) and kicking |
| 10 | F | 3.46 | 50 | 2 | | 582 | 597 | Stepping (T) and kicking |
| 11 | M | 3.47 | 51 | 1 | 90 | 340 | 409 | Kicking |
| 12 | M | 3.39 | 51 | 2 | 122 | 456 | 478 | Stepping (T) and kicking |
| 13 | M | 3.66 | 49 | 2 | 117 | 514 | 536 | Stepping (O) and kicking |
| 14 | M | 3.28 | 48.5 | 21 | 82 | 430 | 446 | Stepping (O) and kicking |
| 15 | F | 3.89 | 51 | 2 | 116 | 413 | 428 | Stepping (O) and kicking |
| 16 | M | 3.55 | 52 | 2 | 120 | 367 | 380 | Stepping (T) and kicking |
| 17 | M | 3.76 | 51 | 8 | 105 | 418 | | Stepping (T) and kicking |
| 18 | F | 3.50 | 49 | 17 | 101 | 515 | | Stepping (O) and kicking |

*et al., 2021*). As *Cheung et al., 2020b* showed that running motor primitives merge with training, we can hypothesize that walking primitives of toddlers will also merge with practice over time. Since learning to walk continues long after walking onset (*Chang et al., 2006*; *Müller et al., 2013*), it is not surprising to notice that the modular organization of toddlers could still need adjustments, even though it suggests that the system might first need to develop the capacity to control muscles more separately before gathering them again into more complex modules. As learning in a modular system relies on both learning the shape of modules and learning their activation parameters (d'*d'Avella and Pai, 2010*), the two processes might not be concomitant to ensure, as suggested above, to always remain in an ideal space of possibilities. Interestingly, recent data from rats report similar modular organization between organisms with different developmental history, suggesting that spinal primitives are determined early in development and conserved into adulthood (*Yang et al., 2019*). However, the fact that the neural repertoire is not mature before the important neural pruning of late adolescence in humans (*de Graaf-Peters and Hadders-Algra, 2006*) suggests that module shaping could continue over a long period of time. The present study falls into this converging framework of long-lasting plasticity of motor modules, associating fractionation and merging of motor primitives throughout development and training (*Bach et al., 2021*; *Cheung et al., 2020b*; *Dominici et al., 2011*; *Hinnekens et al., 2020*; *Sylos-Labini et al., 2020*). While more studies are needed to characterize the maturation of the modular system after walking onset in humans, a long-lasting plasticity of motor primitives could allow to adapt to the development of the musculoskeletal system (*Bizzi et al., 1991*; *Bizzi and Cheung, 2013*) in order to integrate biomechanical specificities of each individual (*Torres-Oviedo and Ting, 2010*) as well as optimality considerations allowing the low-energy costs of mature walking (*Berret et al., 2019*; *Catavitello et al., 2018*; *de Rugy et al., 2012*; *Selinger et al., 2015*).

## Materials and methods

The protocol was in accordance with the Declaration of Helsinki and approved by the French Committee of People Protection. Families were recruited at the Port-Royal maternity in Paris. For each child, a parent provided informed written consent to participate to the study.

### Participants

Eighteen infants (11 males, 7 females) were tested longitudinally from birth to 3 months old, and 15 of them were tested shortly after they could walk independently. Human neonates with a few days of life being a rare clinical population, we chose the number of subjects with the aim of replicating the one of *Teulier et al., 2012*, who analyzed the variability of EMG signal in a longitudinal follow-up of 12 infants between 1 months old and 12 months old in human infants. Inclusion criteria were no known physical or neurological disabilities, gestational age ≥ 38 wk, weight ≥ 2800 g, and APGAR ≥ 8. The walking experiment was set up 1–5 wk after the infant would begin to use walking as the main mode of locomotion, based on parent's reporting by phone call. They were also asked to write down the precise day when the child was able to 'cross an entire room of about 16 feet by walking.' One family forgot to write the precise date but still participated in the walking experiment, and the age of walking onset is given with a precision of ±1 wk (subject ID 11). Three families participated until 3 months only and did not finish the study (one moved away and data collection was stopped for the other two because of the COVID pandemic). Among the remaining toddlers, walking experience was 19.7 ± 14.9 d (mean ± SD) at the time of the experiment. *Table 2* summarizes the ages of the infants at each experiment as well as the behavior that we were able to observe in each infant. The age is not given for S7 and S10 (first and second visits, respectively) because no data could be analyzed from these experiments (either because the child was asleep or because the quality of EMG data was too low; see the procedure for data selection in the section 'Data processing and computed parameters').

### Experimental design

For each experiment, the first stage was to equip the infant or toddler with EMG sensors. Infants were then observed in several positions. At birth and 3 mo, they were observed in a supine position to observe kicking as well as held upright to induce stepping. We also observed two other behaviors that are not reported in this study (crawling and stepping on a treadmill). Each behavior was observed

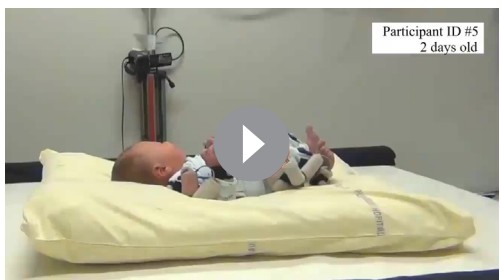

**Video 1.** Video of participant 5 kicking at 2 days old and 86 days old.

https://elifesciences.org/articles/87463/figures#video1

on a pediatric table during about 2 min in a randomized order for each infant and at each age. Shortly after walking onset, toddlers were asked to go back and forth along a 2 m exercise mat. They were walking barefoot at a natural speed during approximately 1 min without any help from adults. Examples of each behavior are shown in *Video 1* (kicking at birth and around 3 months old in participant 5) and *Video 2* (stepping at birth and around 3 months old in participant 6 followed by walking at walking onset).

## Data recording
### EMG recording
Surface EMG data were recorded with the Cometa system at 2000 Hz. At all ages, 10 muscles from shanks, thighs, and buttocks were recorded bilaterally: the tibialis anterior, soleus, rectus femoris, biceps femoris, and gluteus medius. Sensors were also placed on six other muscles of the trunk and the shoulders, but the resulting data were not used in this study. The electrodes placement followed the Surface EMG for Non-Invasive Assessment of Muscles (SENIAM) recommendations (seniam.org).

### Video recording
We used 2D cameras (50 Hz) at each age in order to detect cycles of flexion and extension, synchronized with the EMG recordings. Cameras were placed on each side of the mat or the table so that we would have a clear vision of both sides of the body to detect infants' movements.

## Data processing and computed parameters
### Identification of cycle events
Muscle modules are usually defined as invariant portions of the signal within step cycles. Here usual phases of stance and swing could have been defined for stepping and walking but not for kicking. As we wanted to be able to compare modules from different behaviors, we chose to identify flexion and extension cycles instead of step cycles. Thus, we identified two types of events in each behavior with two trained coders: beginning of hip flexion (BHF; defined as the first frame when hip flexion was elicited) and beginning of hip extension (BHE; defined as the first frame when hip extension was elicited). Reliability of the events identification by the coding procedure was excellent with an intraclass correlation coefficient (ICC) of 0.99 (as done in *Teulier et al., 2012*).

A cycle was defined from a BHF to the following BHF, and made of two phases: the flexion phase, from BHF to BHE, and the extension phase, from BHE to BHF. We always analyzed cycles from the same side of the body, and kept this side for a given baby (for each behavior and each age). We only considered alternated cycles, defined as beginning between 10 and 90% of the cycle of the contralateral lower limb. First and last cycles of an ensemble of alternating cycles were never considered. In order to compare the basic kinematic parameters among behaviors, we computed a few kinematic indexes from this coding of cycles: cycle duration, variability of step duration (standard deviation of step duration divided by the averaged step duration), and proportion of flexion and extension phases. These parameters were chosen because they could be computed in every behavior analyzed in the study (kicking, stepping, and walking).

## Number of cycles included in the analysis
We retained data for our analysis only if a minimum of five alternated cycles of flexion and extension were made and were associated with a clean signal of the 10 muscles simultaneously.

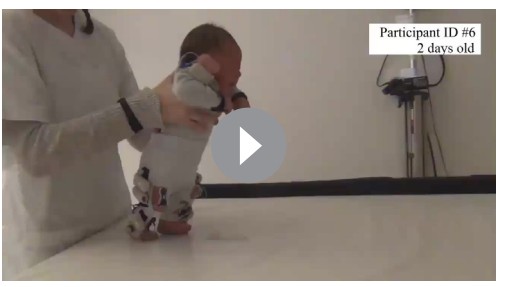

**Video 2.** Video of participant 6 stepping at 2 days old and 74 days old, and walking at 434 days old.

https://elifesciences.org/articles/87463/figures#video2

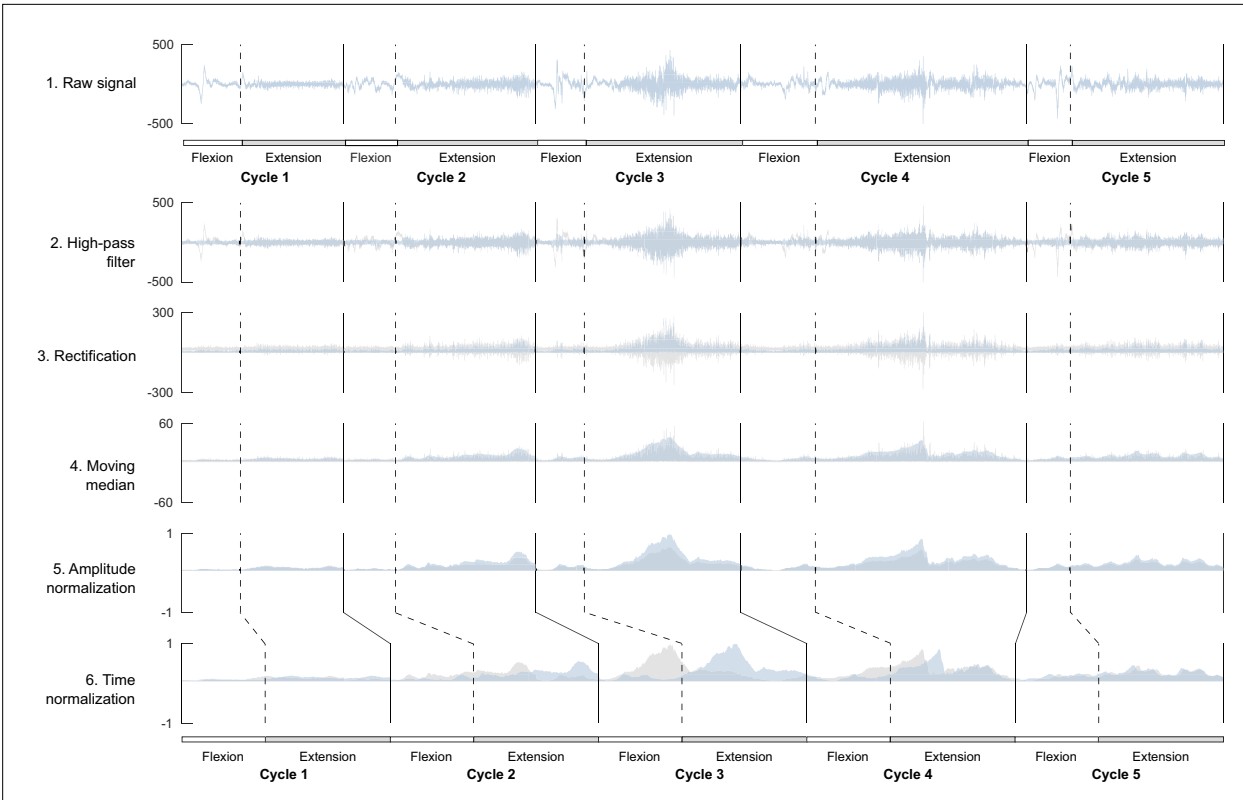

**Figure 6.** Processing of electromyographic (EMG) data preceding factorization. Each line shows one step of the preprocessing for an ensemble of five flexion and extension cycles for one muscle. On each graph, the blue signal represents the named step and the gray signal the previous step (i.e. from the above line). This process is applied to raw data before computing the index of EMG variability and before factorizing the signal to identify motor modules.

When more than five cycles were available, we repeated the analysis on five random combinations of five cycles and averaged the resulting indexes afterward (this allowed to include every step available in the analysis while keeping the indexes of variability comparable across ages and behaviors). Among the 18 infants, 11 performed newborn stepping, 15 performed stepping at 3 months old, 15 performed newborn kicking, 15 performed kicking at 3 months old, and 15 performed walking as toddlers (*Table 2*). A total of 586 cycles were included in the analysis (on average 8.2 per individual).

## EMG processing

EMG data are presented in *Figure 2*, and the procedure for EMG processing is presented in *Figure 6*. For each signal, we applied a high-pass filter (40 Hz, fourth-order Butterworth filter) followed by a rectification, as in *Ivanenko et al., 2013*. To smooth envelopes, we used a moving median since some artifacts were often visible in kicking due to the leg touching the other leg or the mat. The window of this moving median was normalized relatively to the cycle duration that changed with the age of the infant, as described in *Ivanenko et al., 2013*. As cycle duration was about twice lower on average in 3-month-old infants, and again twice lower in toddlers, the moving median window was 400 points in newborns, 200 points in 3 month olds, and 100 points in toddlers. This filtering procedure was applied to every available cycle of each infant and for each age. Signals were normalized for each ensemble of five cycles that were included in the analyses. Finally, cycles were isolated and normalized in time so that each cycle would correspond to 200 time points by interpolating the flexion phase to 80 points and the extension phase to 120 points based on the proportion of phases of independent walking. The rationale here is that all cycles will temporally match the same kinematic events regardless of the age or the behavior. To test how choices on amplitude and time normalization could affect our results, we repeated the analyses of this article with a different amplitude normalization (normalizing under the whole available signals instead of ensembles of five cycles) and with a different time

normalization (interpolating to the cycle without fixing the phases), which confirmed that our results were not dependent of those methodological choices.

As cross-talk might be an issue when recording surface EMG data, we used the same criterion of *Dominici et al., 2011* to assess potential cross-talk (Pearson correlation coefficient >0.2 among pairs of agonist and antagonist muscles). At birth, 3% of our sample had a correlation coefficient >0.2; 1.4% at 3 months, and 6% at walking onset. For these samples, we checked whole recordings and verified that different strides from one subject were not all >0.2.

## Variability of EMG signals

In order to compare the variability of EMG signals across ages, we computed an IEG (*Hinnekens et al., 2020*) from these processed EMG signals as the standard deviation computed point by point across the five cycles. As specified above, when more than five cycles were available, we repeated the analysis on five random combinations of five cycles and averaged the results afterward. This allowed to characterize the motor variability associated with each behavior and at each age before trying to explain how this variability could be generated within a modular system.

## EMG factorization

We extracted spatial and temporal muscle modules into the EMG signals thanks to the space-by-time decomposition method. This method uses NNMF in order to factor EMG signal in a given number of invariant components. As described in *Delis et al., 2014*, this method unifies previous ones as it allows to identify both spatial and temporal EMG modules. In addition, it allows to preserve intra-individual variability into the analysis through activation coefficients that represent the degree of activation of a given module in each cycle. An EMG signal is considered a double linear combination of invariant spatial and temporal modules so that any muscle pattern $m_s(t)$ of the cycle $s$ can be written as

$$m_s(t) = \sum_{i=1}^{P} \sum_{j=1}^{N} w_t(t) a_{i,j,s} w_j + r_s(t)$$

where *P* and *N* are the numbers of temporal and spatial modules, respectively, $w_i(t)$ and $\boldsymbol{w_j}$ are the temporal and spatial modules, respectively, $a_{i,j,s}$ is a scalar activation coefficient (function of the pair of modules it activates and step s), and $r_s(t)$ is the residual reconstruction error describing the difference between the original signal and the reconstructed one. A spatial module is a 10-dimensional vector (as we recorded 10 muscles) and describes ensemble group of muscles that are invariantly activated together across cycles with the same relative proportions. A temporal module is a time-varying function (here described by a 200-dimensional vector) and represents the invariant activation timing of a spatial module within a cycle. The relative shapes of spatial and temporal module are both considered invariant across cycles, but the way pairs of spatial and temporal modules are activated together can differ from one cycle to another. To represent those potentially variable activations, the method computes scalar activation coefficients for each cycle and for each possible pair of spatial and temporal modules, thereby implementing a dimensionality reduction in EMG space. A high activation coefficient corresponds to the concurrent activation of z a specific pair of spatial and temporal modules. The algorithm finds the best fit of spatial and temporal modules and activation coefficients by progressively modifying their values until reaching a convergence criterion. The extraction procedure is repeated 50 times to prevent the risk of finding nonoptimal values because of local minima. This algorithm is detailed in *Delis et al., 2014*. It was run with a custom MATLAB code. Here again, when more than five cycles were available, we repeated this analysis on five random combinations of five cycles and averaged the results afterward (for each index described below). While the goal of NNMF is to model the underlying motor command, we cannot presume about the neural origin of the identified modules, thus we sometimes refer to the identified modular organization as modularity or dimensionality 'within the motor output' in the 'Materials and methods' and 'Results' sections.

## Goodness-of-fit criteria

We computed the VAF as a quality of reconstruction criteria. The VAF is the coefficient of determination between the initial matrix and the reconstructed:

$$VAF = 1 - \frac{\sum_S \|r_s(t)\|^2}{\sum_s \|m_s(t) - \bar{m}\|^2}$$

where $\bar{m}$ is the mean level of muscle activity across all samples and $\|.\|$ represents the Frobenius norm.

The VAF quantifies the goodness of fit between the original EMG patterns and those that are reconstructed from the decomposition. The usual approach is to extract different numbers of modules from each initial matrix and choose the number that allows to get a preset threshold VAF. Nonetheless, it is also interesting to compare the VAF for a given number of module as it allows to directly test the extent to which a low-dimensional command model can give a faithful description of the initial EMG patterns. Moreover, the first approach can only give integers while the second assesses the dimensionality of the motor output by returning reals that might be more precise. Thus we used here both approaches due to their complementarity.

For the first approach, which is to establish a module number given a threshold VAF, we used the threshold of 0.75. The rationale is that numerous studies showed that adults walking could be efficiently modeled by four spatial and temporal modules that are biomechanically functional (*Clark et al., 2010*; *Lacquaniti et al., 2012*; *Neptune et al., 2009*) and that the average VAF of 0.75 was obtained when using the space-by-time decomposition method on nonaveraged signals of adults (*Hinnekens et al., 2020*). Thus we considered this value to represent an efficient modularity with this methodology and used it as a critical threshold. We repeated the extraction and increased the number of spatial and temporal modules until crossing this threshold. For simplicity, we only considered scenarios where the number of spatial modules and the number of temporal modules were equivalent ($P$ = N). This approach allowed us to determine a necessary number of modules in each task and each age. While this number is an integer determined with a threshold, the whole analysis was repeated five times with five random combinations of five cycles, and therefore the number of modules reported in *Figure 3C* can be decimal (see section 'Number of cycles included in the analysis').

For the second approach, which is to establish the VAF for a given number of modules, we again relied on previous results on the EMG decomposition in adults and computed the VAF corresponding to four spatial and temporal modules for each task and each age.

Following those computations, we used the number of modules identified with the first approach to compute the metrics presented below. Nevertheless, those metrics could be influenced by differences in module numbers, thus they were also all computed for four spatial and four temporal modules as a control setting to ensure that significant differences among those metrics would not just be due to differences in the dimensionality of each decomposition (*Supplementary file 1a*, 'Methodological verification').

## Indexes describing the recruitment of modules

We computed two indexes to describe how the recruitment of modules would evolve across ages: IRV and IRS.

The IRV is computed as the average standard deviation of activation coefficients across cycles. As such it quantifies the variability of modules recruitment: a high value means that modules are differently recruited across cycles, while a low value indicates a stable recruitment of modules across cycles.

The IRS corresponds to the sparseness of activation coefficients using a metric described in *Hoyer, 2004*. The sparseness is computed separately for each cycle (and therefore for each vectorized version of a matrix $a_{i,j,s}$, with $i$ denoting its rows and $j$ denoting its columns for the cycle $s$) and averaged afterward. For a given cycle, the resulting value is equal to 1 if the latter vector contains only a single nonzero component, while it is equal to 0 if all components are equal. As such, the IRS indicates the selectivity of activations between spatial and temporal modules regardless of their cycle-to-cycle variability: the larger the IRS is, the more spatial and temporal modules are exclusively paired, while the lower the IRS is, the more spatial and temporal modules are multiplexed.

## Indexes describing the nature of modules themselves

We computed two additional indexes to describe how the nature of modules would evolve across ages: SMAI and STAI.

The SMAI is computed as the average sparseness of spatial modules. Spatial modules are 10-dimensional weighting vectors that can be composed mainly of one muscle (single-muscle module) or 10 muscles with equivalent weights (co-activation module). In general, within a spatial module, muscle will be weighted with more or less. The SMAI quantifies how muscle-selective is a spatial module: when the SMAI is large, it means that spatial modules only gather a few muscles (and in the extreme case, the spatial module is formed with only one muscle), while when it is low, it means that spatial modules gather many different muscles with non-negligible weights.

The STAI is computed as the average sparseness of the temporal activations across temporal modules. The peaks of activations within temporal modules could indeed be narrow (high STAI values), describing refined and precise temporal activation across a cycle, or spread (low STAI values), describing a more continuous activation across the large part of the cycle.

### Computing adult values for comparisons

When analyzing the results, it is often useful to know the value that would be obtained for adults regarding our parameters of interest. Thus we used data from 20 adults from another study (*Hinnekens et al., 2020*) to compute similar indexes on adults. Ten steps were available in each adult, and we used the same approach as in infants, repeating the analysis on five random combinations of five cycles and averaging the resulting indexes afterward. No statistics were made from these data but values are depicted in figures to show the adult landmark (*Figure 1*, black points and gray diamonds in *Figures 3 and 4*, *Supplementary file 1c*).

### Statistical analyses

In addition to reporting the goodness of fit associated with our modeling (i.e. VAF value), we compared features of variability and modularity across ages. Because the number of participants was small, we used nonparametric Wilcoxon tests to compare paired samples across ages from each of the two precursor behaviors to walking on basic kinematic parameters (cycle duration, variability of cycle duration, and proportion of flexion and extension phases across a cycle) and on the variability of EMG signals (IEV). With the same test we analyzed the dimensionality of the motor output by comparing the VAF for a fixed number of modules as well as the number of necessary modules to reach a given VAF. Finally, we compared indexes describing the recruitment of modules (IRV and IRS) as well as indexes describing the nature of modules themselves (SMAI and STAI). Statistical p-values are summarized in *Supplementary file 1a*.

## Acknowledgements

We thank Prof. François Goffinet, head of the Port-Royal maternity in Paris, for encouraging this study. We also thank the Région Ile-de-France for their participation in the initial set-up of the Babylab. Finally, we warmly thank all infants and parents who participated in the study.

## Additional information

### Funding

No external funding was received for this work.

### Author contributions

Elodie Hinnekens, Conceptualization, Software, Formal analysis, Investigation, Visualization, Methodology, Writing - original draft, Project administration; Marianne Barbu-Roth, Conceptualization, Resources, Writing - review and editing; Manh-Cuong Do, Conceptualization, Resources, Supervision, Writing - review and editing; Bastien Berret, Conceptualization, Resources, Software, Formal analysis, Supervision, Methodology, Project administration, Writing - review and editing; Caroline Teulier, Conceptualization, Resources, Formal analysis, Supervision, Investigation, Methodology, Project administration, Writing - review and editing

**Author ORCIDs**
Elodie Hinnekens (iD) http://orcid.org/0000-0003-0649-0779
Caroline Teulier (iD) http://orcid.org/0000-0003-4400-783X

**Ethics**
This study was approved by the French institutional review board "Comité de Protection des Personnes" on November 23, 2017, under the ID RCB 2017-A02596-47. For each child, parents provided informed written consent to participate in the study and publish the study.

**Decision letter and Author response**
Decision letter https://doi.org/10.7554/eLife.87463.sa1
Author response https://doi.org/10.7554/eLife.87463.sa2

---

## Additional files

**Supplementary files**
• Supplementary file 1. Detailed statistics and supplementary individual data. (**a**) Summary of p-values associated with Wilcoxon tests. p-Values are considered significant for p<0.05 (dark gray) and a trend is considered between 0.05 and 0.1 (light gray) E1: birth; E2: 3 months old; E3: walking onset. (**b**) Individual data regarding the dimensionality of the signals: VAF for a modeling of four modules and number of modules to reach the threshold VAF (*Figure 3B*). The goodness of fit is considered sufficient above 75%. Subjects are in the same order than displayed in *Figure 3E*. (**c**) Individual data of 20 adults from *Hinnekens et al., 2020* were used to plot adult landmarks (*Figures 3 and 5*). Note that those data are displayed for illustration: even though we retrieved raw data to apply the same filtering and normalization as in the rest of the article, data are not directly comparable since adult cycles were defined as step cycles (with a stance phase and a swing phase), whereas infant/toddler data were cut off as flexion and extension cycles (with an extension phase and a flexion phase).

• MDAR checklist

• Source code 1. Custom code underlying the computational analysis of this study (sample-based non-negative matrix trifactorization; for more details, see *Delis et al., 2014*).

**Data availability**
Individual data supporting the findings of this study are included in the manuscript, supporting file, and source data files of Figures 2, 3, and 5. The full dataset underlying the computational analysis (EMG data) has been uploaded to https://zenodo.org/record/8193532. The custom code underlying the computational analysis is available as *Source code 1*.

The following dataset was generated:

| Author(s) | Year | Dataset title | Dataset URL | Database and Identifier |
|---|---|---|---|---|
| Hinnekense E, Barbu-Roth M, M-C Do, Berret B, Teulier C | 2023 | Dataset underlying the paper Generating variability from motor primitives during infant locomotor development | https://doi.org/10.5281/zenodo.8193532 | Zenodo, 10.5281/zenodo.8193532 |

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
