## [Editor Report]

This important work on locomotor development takes a longitudinal approach to show that the number of basic locomotor 'primitives' in infant stepping increases from newborn to walking onset, while the variability in their activation decreases. It presents convincing data from the modelling of EMG and kinematic data, which should be of interest to physiologists and psychologists interested in motor skills and development.

---

## [Decision Letter]

**Decision letter after peer review:**

[Editors’ note: the authors submitted for reconsideration following the decision after peer review. What follows is the decision letter after the first round of review.]

Thank you for submitting the paper "Generating variability from motor primitives during infant locomotor development" for consideration by *eLife*. Your article has been reviewed by 3 peer reviewers, and the evaluation has been overseen by a Reviewing Editor and a Senior Editor. The following individual involved in the review of your submission has agreed to reveal their identity: Vincent C. K. Cheung (Reviewer #2).

Comments to the Authors:

We are sorry to say that, after consultation with the reviewers, we have decided that this work will not be considered further for publication by *eLife* at this time.

Specifically, while we very much appreciated the difficulty of infant longitudinal data collection, we felt that the dataset presented here was not large and consistent enough to warrant its conclusions. The fact that analyses were split into steps and kicks, with low numbers of participants in each, did not allow R1 or R3 to feel confident about its publication as it stands. Both R1 and R3 also make the point that in such a rich dataset it is not clear why certain cycles, samples or measures were selected. Finally, R2 notes that an alternative interpretation of your results might be possible – rejecting (or accepting) this would require some remodelling of your data. We all felt that the theoretical point of your paper was important, and we welcomed your general approach. We are therefore sorry not to be able to recommend it for publication at this time. However, if you feel that you can address these issues, we are open to a resubmission. In that case, we would ask you to include a point-by-point response to the reviews here, detailing the changes you have made. In any event we commend you on this interesting work and wish you all the best with future submissions.

*Reviewer #1 (Recommendations for the authors):*

In terms of introducing and interpreting the work, and situating your study in light of others' and in terms of children's overall behavioural repertoire, I think this could be done by additions to the text. I will add here that although I am not familiar with it, it seems that the paper by Sylos-labini (2020) is very relevant, and I think it would be good to make more explicit the similarities and differences between your studies.

My point about analyses being based on a limited sample of data in the Public Review, is for me the major weakness of the study, and I think the very simple solution is to collect more data. I think if you could show the patterns you do in a larger sample, the paper would make a far more substantial contribution to the literature. I really enjoyed the general approach you have taken, but I just don't feel convinced that you can draw strong conclusions from the current dataset. To expand a little on this point: I know that from Thelen's work, early kicking movements are kinematically very similar to steps, but I question whether you can really select these as entirely equivalent. Indeed Figure 2 demonstrates clear differences in the EMG patterns of steps and kicks. If these are not to be considered equivalent, then as far as I understand, the data on the development of stepping is based on a sample of n=6, which is really very low, and I do not think is enough to draw the broad conclusions that one would want for this journal. Likewise, if 6/12 contributed steps at birth and 9/12 contributed kicks, it would be useful to know how much these groups overlapped and what the similarities were between a single infant's steps and kicks.

Finally, I have some additional points on presentation which I hope are helpful for future submissions:

Figure 1

It's not clear to me how the left-right alternating gait pattern comes into play in the model. From the spatial module plots, it looks like there is no particular correlation between a left-foot step and the subsequent right-foot step. Is that true? Is the temporal structure of the gait cycle imposed on the variability in any way or included in the model? Relatedly, why are the spatial but not the temporal modules split by leg?

Figure 2

In Figure 2, from the legend it's not clear why only high-pass filtered data is shown, why the reader needs to know this, and when low-pass filters were also applied. Figure 2 legend Typo: "The scale if 1 second is displayed"

Since the topic is variability, it would be good to see not just the full patterns across 10 muscles for each behaviour/ timepoint, but e.g. indicative traces representing variability in 1 muscle at early and late time points.

*Reviewer #2 (Recommendations for the authors):*

The authors' argument can potentially be made a lot stronger if alternative models of motor modules that permit cycle-to-cycle variability of the spatial and temporal modules themselves can be considered. For instance, with the NMF algorithm, it is possible to extract trial-specific w_i_(t) and w_j_ with a second extraction, using results of the first extraction as initial estimates (e.g., see Cheung et al., 2020, IEEE-OJEMB; but there may be other better methods). It can then be assessed whether the variance of w_i_(t), a_ijs_, or w_j_ relates the best to the kinematic and/or EMG variability.

*Reviewer #3 (Recommendations for the authors):*

I recommend the authors will collect data from more individuals, in more data points while being consistent with the recording timing. The paper would also benefit from more clarification about the motivation for the study and the rationale of the measurements and the specific techniques. Authors should also consider recording data in the natural environment.

After collecting the additional data, I recommend that the authors will conduct analyses that focus on individual differences that will use the power of longitudinal recordings to provide insights into the development of motor primitives (that can be using unsupervised machine learning, or any other approach). Finally, to cope with infants' real-life changes, I strongly recommend analysing motor movement more generally without focusing on infants' alternating leg movements.

[Editors’ note: further revisions were suggested prior to acceptance, as described below.]

Thank you for submitting your article "Generating variability from motor primitives during infant locomotor development" for consideration by *eLife*. Your article has been reviewed by 2 peer reviewers, and the evaluation has been overseen by a Reviewing Editor and Tamar Makin as the Senior Editor.

In general, the reviewers were happy with your revisions, but they have made some relatively minor further suggestions for improving the clarity and presentation of the paper, we would be grateful if you could consider the reviewers suggestions appended below (both public and for authors) while revising your manuscript.

*Reviewer #1 (Recommendations for the authors):*

As reviewer I made 4 main points.

1. Introduce neural and developmental aspects of the work; relate to more varied patterns of walking.

The reviewer has now added these points in the introduction, giving the reader better context for understanding the work. Still, I suggest that the extant developmental literature on learning to walk needs better integrated into the ms – an excellent place to start is Adolph and Cole, TICS 2018.

2. The dataset is limited and sparse.

The work is now based on a greater number of participants (18 vs 10) and a greater number of cycles (586 vs 200). This enhances the robustness of the work. It is apparent that many of the same patterns apparent in the earlier, more stringent version of the analysis, are still present in this broader dataset. The inclusion of treadmill steps and non-consecutive steps does not seem unreasonable. The table is helpful.

3. Interpretation is unclear at points.

In their rebuttal the authors give an interesting comment on purposeful vs incidental variability. There is no need to go into this in the paper in more detail. You now clarify the difference from the Sylos-Labini paper better. On re-reading Dominici Science, you should clarify why your absolute number of modules is higher than theirs (e.g. yours 4 vs Domini 2 at birth?)

4. You need to clarify aspects of the Figures

These points are now clearer.

*Reviewer #2 (Recommendations for the authors):*

I read this revised version in detail and found it to be more compelling than the previous version. The Introduction, in particular, is a lot more well written. As shown in your new analysis provided in the rebuttal, it is reassuring that the trend of a decreasing module recruitment variability from neonates to toddlers could still be observed even when the modules themselves were permitted to vary cycle-by-cycle, thus indirectly suggesting that the changes of variability across stages is likely channeled, at least in part, through activations of the modules rather than variability of the modules themselves. I suggest putting this new analysis as supplementary information (if possible) for the interested readers, so that the overall readability of the main text can be kept.

---

## [Author Response]

[Editors’ note: the authors resubmitted a revised version of the paper for consideration. What follows is the authors’ response to the first round of review.]

Comments to the Authors:We are sorry to say that, after consultation with the reviewers, we have decided that this work will not be considered further for publication by eLife at this time.Specifically, while we very much appreciated the difficulty of infant longitudinal data collection, we felt that the dataset presented here was not large and consistent enough to warrant its conclusions. The fact that analyses were split into steps and kicks, with low numbers of participants in each, did not allow R1 or R3 to feel confident about its publication as it stands. Both R1 and R3 also make the point that in such a rich dataset it is not clear why certain cycles, samples or measures were selected. Finally, R2 notes that an alternative interpretation of your results might be possible – rejecting (or accepting) this would require some remodelling of your data. We all felt that the theoretical point of your paper was important, and we welcomed your general approach. We are therefore sorry not to be able to recommend it for publication at this time. However, if you feel that you can address these issues, we are open to a resubmission. In that case, we would ask you to include a point-by-point response to the reviews here, detailing the changes you have made. In any event we commend you on this interesting work and wish you all the best with future submissions.

We thank the editors and reviewers for their deep assessment of our work as it really helped us to improve our paper. Here is a summary of the major additions and modifications that were made:

1) The number of participants included in the analysis was significantly increased In the previous version of the manuscript, we presented data on 12 infants, of which 6 had stepped, 9 had kicked and 10 had walked. While these data were particularly hard to get (because at first newborn could stay awake only a limited time with a restrictive skin area to add adhesive on, then because of the challenge of the longitudinal follow-up), the limited aspect of the cohort was also due to the fact that we chose a very conservative approach and did not present our whole dataset. In the previous version of the manuscript, a behavior was considered analyzable when the infant electromyographic (EMG) recordings were available on all 10 muscles during 5 flexion and extension cycles of the lower-limb that were selected among sets of at least 3 consecutive cycles. We did not include infants who stepped on a treadmill instead of overground, and we only included infants who were moving during the majority of the trial. Furthermore, these conditions had to be met within a same behavior on at least the first two time points of the longitudinal follow-up (birth and 3 months old).

Following the reviewers’ comments, we understood that we were too conservative and revised these criteria in light of our hypothesis and of the current literature. Mainly, we now analyze behavior in infants who moved in a less stereotypical manner (i.e. even if they did not move during the majority of the trial, as long as they produced alternated flexion and extension cycles of the lower-limbs and even if these cycles were not consecutive). We also included infants who stepped on a treadmill, as it is typically done in the literature on newborn stepping (Sylos-Labini et al., 2022; Sylos-labini et al., 2020). Therefore, we now present data on our whole cohort, i.e. on 18 infants who were longitudinally followed from birth on, of which 11 stepped at birth, 15 stepped at 3 months old, 15 kicked at birth, 15 kicked at 3 months old, and 15 were recorded at walking onset. Data are still exclusively longitudinal in the current version (we present data for a given child if they are available on at least two time points).

2) The number of steps included in the analysis was significantly raised

In the previous version of the manuscript, we had fixed the number of cycles to analyze in each participant, thereby limiting the number of steps included in the analysis to 5 per individual. The rationale was that the analysis focused on intra-individual variability, and therefore fixing the number of data made this variability comparable across ages and individuals. For example, to compute the Index of EMG Variability (IEV), EMG data from each cycle of flexion and extension is normalized to 200 time points, and a standard deviation is computed for each of these time points across the 5 cycles that are included in the analysis. The IEV is the sum of all of these standard deviations, across every time point and every muscle. As standard deviation estimates may depend on the initial number of data, we considered that fixing the number of analyzed cycles to five was a conservative way to make intra-individual variability strictly comparable across individuals and ages. In the same vein, the extraction of muscle synergies depends on the number of cycles included in the analysis (Oliveira et al., 2014), which also encouraged us to standardize the number of cycles across individuals and ages. As such, the number of 5 cycles was previously chosen to match the minimal number of cycles that was available in some infants, and when more cycles were available, we randomly selected the 5 cycles to analyze. In the new version of the manuscript, we preprocessed data for every available cycle in every individual, and when more than 5 cycles were available, we repeated our analysis for 5 random combinations of 5 cycles that were selected out of all available ones (i.e. we used a bootstrapping approach, limiting the number of combinations to 5 because of the processing time of the algorithm). This allowed to increase. The number of cycles included in the analysis and therefore to improve the reliability of our estimations while keeping the indexes of variability based on 5 cycles and comparable across individual and ages. The results that we present are thus the averaged result across these five combinations. Thanks to this new approach, a total of 586 cycles are now included in the analysis (on average 8.2 per individual) compared to 200 cycles in the previous version. The detailed number for each individual and ages is presented in Author response table 1. This procedure was detailed in the method section (line 584-595)

**Author response table 1. sa2table1:** Overview of data included in the analysis.

Participant ID	Overview of data included in the analysis
Stepping	Walking onset	Kicking
Birth	3 months old	3 months old	Birth
1	8	5	8	11	6
2	5	9	22	5	
3	7	5	8	5	
4		7	5	8	6
5				12	7
6	6	5	7		8
7		16	9	6	
8	6	6	12	5	5
9	6	8	22	12	8
10	6		13		6
11			7	5	6
12		9	18	10	5
13	5	5	20	6	6
14		6	5	7	9
15		8	7		8
16	15	11	7	8	5
17	9	13		7	5
18	11	9		8	5
Total number of participants included in the analysis	11	15	15	15	15
Total number of steps included in the analysis	84	122	170	115	95

3) We clarified the text on several aspects, including the type of variability that we are investigating and how it relates to other levels of variability during development (line 34-44). We also explained in more details how this paper was different from other paper on infants’ modularity (line 72-96) and added a table recapitulating the role of the main variables of the study (Table 1 in the ms, line 176). In the discussion, we clearly stated that the current study did not allow to discriminate among purposeful and incidental variability and that other studies are needed to explore the extent of our results within other early behaviors and coordination modes (line 388-395).

4) We remodeled our data as suggested and verified our main result under the assumption that modules could be plastic on the short term (see details of the analysis in response to reviewer 2).

Reviewer #1 (Recommendations for the authors):In terms of introducing and interpreting the work, and situating your study in light of others' and in terms of children's overall behavioural repertoire, I think this could be done by additions to the text. I will add here that although I am not familiar with it, it seems that the paper by Sylos-labini (2020) is very relevant, and I think it would be good to make more explicit the similarities and differences between your studies.

As detailed above we reviewed the introduction to be clearer about the type of variability that we focus on. We also clarified how the paper of Sylos-Labini was different from our study (this paper does not focus on variability because the modelling is based on single-step data, but rather focusses on comparing stepping and kicking as two distinct locomotor precursors, see line 77-78 and 88-89).

My point about analyses being based on a limited sample of data in the Public Review, is for me the major weakness of the study, and I think the very simple solution is to collect more data. I think if you could show the patterns you do in a larger sample, the paper would make a far more substantial contribution to the literature. I really enjoyed the general approach you have taken, but I just don't feel convinced that you can draw strong conclusions from the current dataset. To expand a little on this point: I know that from Thelen's work, early kicking movements are kinematically very similar to steps, but I question whether you can really select these as entirely equivalent. Indeed Figure 2 demonstrates clear differences in the EMG patterns of steps and kicks. If these are not to be considered equivalent, then as far as I understand, the data on the development of stepping is based on a sample of n=6, which is really very low, and I do not think is enough to draw the broad conclusions that one would want for this journal. Likewise, if 6/12 contributed steps at birth and 9/12 contributed kicks, it would be useful to know how much these groups overlapped and what the similarities were between a single infant's steps and kicks.

As explained above, we significantly raised the number of analyzed data: while the previous version of the manuscript was based on 200 cycles that were observed in 12 individuals, we now present 586 cycles that were observed in 18 individuals, 15 of which were followed until walking onset. Moreover, the extent to which the groups overlap is detailed in Table 2 (method section, line 540). While we agree with the reviewer that assessing the similarities between steps and kicks would be very interesting, this was done by other authors (Sylos-labini et al., 2020) and we relied on their results to repeat our analysis in what appear to be two distinct locomotor precursors. As also mentioned above, we clarified the range of the paper that focusses on one scale of variability (i.e. the intra-individual variability that can be found within a given behavior). As such we feel that comparing two behaviors would answer a too different question and should constitute another paper.

Finally, I have some additional points on presentation which I hope are helpful for future submissions:Figure 1It's not clear to me how the left-right alternating gait pattern comes into play in the model. From the spatial module plots, it looks like there is no particular correlation between a left-foot step and the subsequent right-foot step. Is that true? Is the temporal structure of the gait cycle imposed on the variability in any way or included in the model? Relatedly, why are the spatial but not the temporal modules split by leg?

The legend was indeed not clear enough on one point: the modeling is based on the assumption that motor modules are bilateral (as was the case in similar literature, see Dominici et al., 2011; Syloslabini et al., 2020). Therefore, kinematic events are defined on one limb only (here the right one) but muscle from both sides are recorded and assumed to be controlled together (which do not mean that the algorithm forces modules to be bilateral – if they were not any correlation across muscles from different sides the algorithm would find unilateral modules – but only that muscles can be gathered in a given spatial module regardless on their original side). The legend of the figure was clarified (line 136-138).

Figure 2In Figure 2, from the legend it's not clear why only high-pass filtered data is shown, why the reader needs to know this, and when low-pass filters were also applied. Figure 2 legend Typo: "The scale if 1 second is displayed"Since the topic is variability, it would be good to see not just the full patterns across 10 muscles for each behaviour/ timepoint, but e.g. indicative traces representing variability in 1 muscle at early and late time points.

Thanks for the suggestion, we initially plotted only high-pass filtered data because this filter mainly imply a straightening of the baseline and therefore allows a clear representation. However, we indeed realized that a clear overview of intra-individual variability was lacking in this Figure. Therefore Figure 2 was modified and we now present, in addition to a few high-pass filtered data, completely preprocessed data (i.e. filtered and normalized) that are superimposed with the corresponding averaged signal in order to depict the variability. The whole process of filtering also appears in figure 6, and to clarify the effect of high-pass filtering we now detail this step in the figure.

Reviewer #2 (Recommendations for the authors):The authors' argument can potentially be made a lot stronger if alternative models of motor modules that permit cycle-to-cycle variability of the spatial and temporal modules themselves can be considered. For instance, with the NMF algorithm, it is possible to extract trial-specific w_i_(t) and w_j_ with a second extraction, using results of the first extraction as initial estimates (e.g., see Cheung et al., 2020, IEEE-OJEMB; but there may be other better methods). It can then be assessed whether the variance of w_i_(t), a_ijs_, or w_j_ relates the best to the kinematic and/or EMG variability.

Thanks for the suggestion. We agree with the reviewer than the only variable that is adjustable for explaining motor variability is the modules’ activation coefficients. However, we want to stress that our conclusions are not only based on the fact that activation coefficients vary but on the fact that their variability evolves with the dimension of modularity (i.e. while we applied the same model to all of our data, variability is better explained by highly variable activations of a few number of modules in early life, whereas it is better explained by steadier activations of a high number of modules in toddlerhood). Doing so, we implicitly assumed that modules cannot vary from cycle to cycle (very short time scale) but that they can vary across development (long time scale). This is the common assumption in muscle synergy analyses (e.g., d’Avella et al., 2003). Nevertheless, to test whether our findings hold if we allow modules to vary across cycles, we reproduced our analysis with the method presented in Cheung et al. (2020). Here the method was adapted to our space-by-time model by extracting trial-specific w_is_(t) and w_js_ using the global/fixed w_i_(t) and w_j_ as initial estimates of the iterative algorithm. Under this hypothesis we estimated the short-term plasticity of spatial and temporal modules by computing the sum of point-by-point standard errors across modules from different cycles. An illustration of this analysis is reported in the Author response image 1 and 2:

**Author response image 1. sa2fig1:** 

This figure illustrates the modules obtained with the plastic-modules approach: the algorithm was initialized with modules that were identified through the original approach and identified different modules for each of the five cycles.

We computed the plasticity of spatial modules, of temporal modules and of activation coefficients (i.e. the IRC) for the specific number of modules of each individual and age (A, B, C) and for a number of modules fixed to 4 (D, E, F). In any case, the main effect of the paper (decrease of the IRC with age) was persistent, indicating that the algorithm explains better the variability of newborn data by the variability of activation coefficients than for toddlers’ data even under the hypothesis that modules could be plastic in the short-term (i.e. cycle-to-cycle). We repeated this analysis with or without initiating the activation coefficients, and with or without gathering modules by best matching pairs afterwards, and found similar results. This control analysis, showing a good robustness of our main findings, is now briefly mentioned in the paper (line 277-279). We chose not to provide more details in the main text for overall readability of the paper.

Reviewer #3 (Recommendations for the authors):I recommend the authors will collect data from more individuals, in more data points while being consistent with the recording timing. The paper would also benefit from more clarification about the motivation for the study and the rationale of the measurements and the specific techniques. Authors should also consider recording data in the natural environment.After collecting the additional data, I recommend that the authors will conduct analyses that focus on individual differences that will use the power of longitudinal recordings to provide insights into the development of motor primitives (that can be using unsupervised machine learning, or any other approach). Finally, to cope with infants' real-life changes, I strongly recommend analysing motor movement more generally without focusing on infants' alternating leg movements.

As explained above, we were able to significantly increase the cohort of infants that was presented in this study. We also added several precisions regarding the motivation for the study and the related measurements and techniques. Moreover, we fully agree with the reviewer than recording data in the natural environment is extremely impactful in developmental sciences, however our analyses focused on the ability to generate a given movement in different ways, and therefore that it was methodologically pertinent in this case to fix other sources of variability (and to focus on one coordination. mode). Finally, as also explained above, we included a new paragraph in the discussion to discuss inter-individual difference.

Overall, as we presented a new approach, we fully agree that this study is associated with a number of limits that open the numerous perspectives that were suggested by the reviewer (e.g. to study how variability interacts with modularity within other behaviors, other ages and other environment, and to understand how other variables might interact with the one that we introduced) and that we cited within the discussion.

References

Chang C-L, Kubo M, Buzzi U, Ulrich BD. 2006. Early changes in muscle activation patterns of toddlers during walking. Infant Behav Dev 29:175–88. doi:10.1016/j.infbeh.2005.10.001

Cheung VCK, Zheng XC, Cheung RTH, Chan RHM. 2020. Modulating the structure of motor variability for skill learning through specific muscle synergies in elderlies and young adults. IEEE Open J Eng Med Biol 1:33–40. doi:10.1109/OJEMB.2019.2963666

d’Avella A, Saltiel P, Bizzi E. 2003. Combinations of muscle synergies in the construction of a natural motor behavior. Nat Neurosci 6:300–308. doi:10.1038/nn1010

Dhawale AK, Smith MA, Ölveczky BP. 2017. The Role of Variability in Motor Learning. Annu Rev Neurosci 40:479–498. doi:10.1146/annurev-neuro-072116-031548

Dominici N, Ivanenko YP, Cappellini G, D’Avella A, Mondì V, Cicchese M, Fabiano A, Silei T, di Paolo A, Giannini C, Poppele RE, Lacquaniti F. 2011. Locomotor primitives in newborn babies and their development. Science 334:997–9. doi:10.1126/science.1210617

Hoyer PO. 2004. Non-negative Matrix Factorization with Sparseness Constraints. Journal of Machine Learning Research 5:1457–1469.

Kiehn O, Kjærulff O. 1996. Spatiotemporal characteristics of 5-HT and dopamine-induced rhythmic hindlimb activity in the in vitro neonatal rat. J Neurophysiol 75:1472–1482. doi:10.1152/jn.1996.75.4.1472

Klein DA, Patino A, Tresch MC. 2010. Flexibility of Motor Pattern Generation Across Stimulation Conditions by the Neonatal Rat Spinal Cord. J Neurophysiol 103:1580–1590. doi:10.1152/jn.00961.2009.

Martorell R, de Onis M, Martines J, Black M, Anyango A, Dewey K. 2006. WHO Motor Development Study: Windows of achievement for six gross motor development milestones. Acta Paediatrica, International Journal of Paediatrics 95:86–95. doi:10.1080/08035320500495563

Oliveira AS, Gizzi L, Farina D, Kersting UG, Clark DJ, Va MR. 2014. Motor modules of human locomotion : influence of EMG averaging , concatenation , and number of step cycles. Front Hum Neurosci 8:1–9. doi:10.3389/fnhum.2014.00335

Sylos-Labini F, la Scaleia V, Cappellini G, Dewolf A, Fabiano A, Solopova IA, Mondì V, Ivanenko Y, Lacquaniti F. 2022. Complexity of modular neuromuscular control increases and variability decreases during human locomotor development. Commun Biol 5:1256. doi:10.1038/s42003022-04225-8

Sylos-labini F, Scaleia V la, Cappellini G, Fabiano A, Picone S, Keshishian ES, Zhvansky DS, Paolillo P, Irina A, D’Avella A, Ivanenko YP, Lacquaniti F, Solopova IA, D’Avella A, Ivanenko YP, Lacquaniti F. 2020. Distinct locomotor precursors in newborn babies. Proceedings of the National Academy of Sciences 117:9604–9612. doi:10.1073/pnas.1920984117

Teulier C, Sansom JK, Muraszko K, Ulrich BD. 2012. Longitudinal changes in muscle activity during infants’ treadmill stepping. J Neurophysiol 108:853–862. doi:10.1152/jn.01037.2011

[Editors’ note: what follows is the authors’ response to the second round of review.]

Reviewer #1 (Recommendations for the authors):As reviewer I made 4 main points.1. Introduce neural and developmental aspects of the work; relate to more varied patterns of walking.The reviewer has now added these points in the introduction, giving the reader better context for understanding the work. Still, I suggest that the extant developmental literature on learning to walk needs better integrated into the ms – an excellent place to start is Adolph and Cole, TICS 2018.

Thanks for the suggestion, we integrated several considerations of this paper is the manuscript. In particular we developed several aspects of the introduction and of the discussion, see below what we added in bold:

Line 34, first paragraph of the introduction: “Variability arises at several levels of the motor system during early locomotor development. Firstly, as soon as birth, infants are able to perform a wide range of behaviors involving flexion and extension cycles of the lower-limbs, such as stepping, kicking swimming or crawling (Forma et al. 2019, McGraw, 1941a, 1939; Sylos-Labini et al., 2020; Thelen and Fisher, 1983, 1982). Secondly, a given behavior can be realized with numerous coordination modes. For example, neonatal stepping can involve alternated steps, parallel steps, serial steps or single steps (Siekerman et al., 2015). Similarly, toddlers can follow curved paths when walking or generate a variety of coordination patterns on the fly when cruising over varying distances (Ossmy and Adolph, 2020). Thirdly, a given coordination mode can be realized by different combinations of muscles. For example, infants demonstrate a high variability of muscle activations throughout their first year of life when stepping or kicking, even when producing only alternated leg movements (Sylos-Labini et al., 2020; Teulier et al., 2012). This multi-level variability can arise in numerous environmental contexts and is associated with the development of multiple components, like the growth of musculoskeletal structures, the myelination of neural circuits, or the motivational goal to move, leading infants to learn new skills with their own developmental time scale (Adolph et al. 2018).”

Line 368 “As we observed such structure within alternated leg movements, other studies are needed to explore the extent of these results to other early behaviors or coordination modes, as well as their link with the ability to move in various environmental contexts.”

Line 446: “Overall, when compared with adult values (Figure 3, Figure 5, supplementary file 1a), our results suggest an immaturity of the modular system before and around walking onset, which confirms that infancy should be an ideal period of plasticity to benefit from in therapy (Ulrich et al., 2010; Morgan et al., 2021). There are also coherent with the idea that skill onset during development is not an on-off switch and that maturity is not reached as soon as a skill becomes possible (Adolph et al. 2018).”

2. The dataset is limited and sparse.The work is now based on a greater number of participants (18 vs 10) and a greater number of cycles (586 vs 200). This enhances the robustness of the work. It is apparent that many of the same patterns apparent in the earlier, more stringent version of the analysis, are still present in this broader dataset. The inclusion of treadmill steps and non-consecutive steps does not seem unreasonable. The table is helpful.3. Interpretation is unclear at points.In their rebuttal the authors give an interesting comment on purposeful vs incidental variability. There is no need to go into this in the paper in more detail. You now clarify the difference from the Sylos-Labini paper better. On re-reading Dominici Science, you should clarify why your absolute number of modules is higher than theirs (e.g. yours 4 vs Domini 2 at birth?)

The absolute number of modules is often defined by setting a VAF threshold but there is no consensus about this threshold, and therefore this number is always arbitrary. As such, it makes more sense to compare the evolution of the number of modules than the absolute number (we indeed observe an increase in the number of modules between birth and independent walking as in Dominici et al. 2011). This appears in the discussion line 425: “When working with EMG to identify hypothetical motor primitives, several factors can limit interpretations such as EMG processing, cross-talk, or arbitrary choices made during EMG factorization. It is indeed worth noticing that the current study report different values than previous developmental studies regarding the absolute number of modules (Dominici et al., 2011; Sylos-labini et al., 2020). However, there is currently no consensus regarding the selection of the number of modules which depends on arbitrary criteria such as the VAF threshold.”

Reviewer #2 (Recommendations for the authors):I read this revised version in detail and found it to be more compelling than the previous version. The Introduction, in particular, is a lot more well written. As shown in your new analysis provided in the rebuttal, it is reassuring that the trend of a decreasing module recruitment variability from neonates to toddlers could still be observed even when the modules themselves were permitted to vary cycle-by-cycle, thus indirectly suggesting that the changes of variability across stages is likely channeled, at least in part, through activations of the modules rather than variability of the modules themselves. I suggest putting this new analysis as supplementary information (if possible) for the interested readers, so that the overall readability of the main text can be kept.

Thanks for the suggestion, we added this supplementary analysis in the appendix as two supplementary figures (Figure 4—figure supplement 1 and Figure 4—figure supplement 2).